# Modulation of fungal virulence through CRZ1 regulated F-BAR-dependent actin remodeling and endocytosis in chickpea infecting phytopathogen *Ascochyta rabiei*

**Manisha Sinha**[1☉], **Ankita Shree**[1☉], **Kunal Singh**[1¤], **Kamal Kumar**[1], **Shreenivas Kumar Singh**[1], **Vimlesh Kumar**[2], **Praveen Kumar Verma**[1,3]*

**1** Plant Immunity Laboratory, National Institute of Plant Genome Research, Aruna Asaf Ali Marg, New Delhi, India, **2** Department of Biological Sciences, Indian Institute of Science Education and Research Bhopal (IISER-Bhopal), Bhauri, Bhopal, India, **3** Plant Immunity Laboratory, School of Life Sciences, Jawaharlal Nehru University, New Delhi, India

☉ These authors contributed equally to this work.
¤ Current address: Biotechnology Division, CSIR-Institute of Himalayan Bioresource Technology, Palampur, India
* pkv@nipgr.ac.in, praveenkverma@jnu.ac.in

**Data Availability Statement:** All relevant data are within the manuscript and its Supporting Information files.

## Abstract

Polarized hyphal growth of filamentous pathogenic fungi is an essential event for host penetration and colonization. The long-range early endosomal trafficking during hyphal growth is crucial for nutrient uptake, sensing of host-specific cues, and regulation of effector production. Bin1/Amphiphysin/Rvs167 (BAR) domain-containing proteins mediate fundamental cellular processes, including membrane remodeling and endocytosis. Here, we identified a F-BAR domain protein (ArF-BAR) in the necrotrophic fungus *Ascochyta rabiei* and demonstrate its involvement in endosome-dependent fungal virulence on the host plant *Cicer arietinum*. We show that ArF-BAR regulates endocytosis at the hyphal tip, localizes to the early endosomes, and is involved in actin dynamics. Functional studies involving gene knockout and complementation experiments reveal that ArF-BAR is necessary for virulence. The loss-of-function of *ArF-BAR* gene results in delayed formation of apical septum in fungal cells near growing hyphal tip that is crucial for host penetration, and impaired secretion of a candidate effector having secretory signal peptide for translocation across the endoplasmic reticulum membrane. The mRNA transcripts of *ArF-BAR* were induced in response to oxidative stress and infection. We also show that ArF-BAR is able to tubulate synthetic liposomes, suggesting the functional role of F-BAR domain in membrane tubule formation *in vivo*. Further, our studies identified a stress-induced transcription factor, ArCRZ1 (Calcineurin-responsive zinc finger 1), as key transcriptional regulator of *ArF-BAR* expression. We propose a model in which ArCRZ1 functions upstream of *ArF-BAR* to regulate *A. rabiei* virulence through a mechanism that involves endocytosis, effector secretion, and actin cytoskeleton regulation.

**Funding:** This research received funding from the Core Grant of National Institute of Plant Genome research to PKV., Department of Biotechnology, Government of India, Grant number: BT/PR10605/PBD/16/791/2008 to PKV, Department of Biotechnology, Government of India, Grant number: BT/AGR/CG-Phase II/01/2014 to PKV and Council of Scientific and Industrial Research, Government of India, Grant number: 19/06/2016(i) EU-V to AS. The funders had no role in study design, data collection and analysis, decision to publish, or preparation of the manuscript.

**Competing interests:** The authors have declared that no competing interests exist.

## Author summary

BAR-domain superfamily is known to mold amorphous lipid bilayer into defined tubular shapes and required for endosome formation and trafficking. Although these proteins were studied earlier in context of their structural and biochemical properties, there is limited evidence on the direct role of F-BAR domain proteins in the pathophysiological development of other economically important fungi. Our study assumes functional significance for plant infection as we identified an F-BAR domain-containing protein that is regulated by a distinct transcriptional network regulated by a calcium-regulated CRZ1 transcription factor. We characterized ArF-BAR in a necrotrophic fungal pathogen, *Ascochyta rabiei*, which causes the Ascochyta blight disease of chickpea. Our study will help to understand a signaling cascade that regulates the formation of endosomes, which is required for fungal virulence.

## Introduction

Polarized hyphal growth is a signature feature of filamentous fungi during host colonization [1]. This feature allows the fungus to sense, coordinate, and respond to an array of cues from the host [2]. Therefore, regulation of hyphal tip growth is one of the major virulence determinants in filamentous fungi. In response to infection, the plant innate immune system recognizes pathogen and initiate effective defense response. Pathogens must recognize the plant surface cues and counter host-generated defense responses for effective pathogenesis. Additionally, invasive fungi must overcome an intracellular challenge posed by the distance between the elongated invading hyphal architecture and the nucleus [3]. Mounting evidence strongly suggests that the complications associated with this increased distance are overcome by long-distance intracellular communication for rapid and precise transduction of external information [4]. Moreover, the maintenance of extremely polarized hyphal morphology is heavily dependent on endosome trafficking. It is also important to maintain the structural and functional features of a fungal cell [5].

In the case of filamentous fungi, long-distance signaling is mediated by endosomes, which are multipurpose carriers. Besides signal sensing for motor-dependent retrograde signaling, endosomes are involved in the recycling of cell wall components, polarisome proteins, and various receptors required for polarized tip growth [6]. Loss of these functions leads to impaired host invasion and virulence [7,8]. The generation of early endosomes (EEs) is a key step in the endocytic pathway and involves the cooperative action of membrane bending and cytoskeleton reorganization. Membrane bending is the cornerstone for the generation of EEs and is regulated by proteins involved in the detection and stabilization of membrane curvature [9,10].

In animal cells, BAR domain superfamily proteins have been shown to regulate membrane dynamics and remodelling of the actin cytoskeleton [11]. The F-BAR domain proteins form N-terminal α-helical coiled-coil dimers and bind to negatively charged lipid membranes via their positively charged residues. This binding generates membrane curvature and regulates intracellular vesicle trafficking [12]. Depending on the degree of curvature in each dimer of BAR proteins, the BAR domain superfamily proteins are broadly classified into three families: classical BAR and N-BAR, Fer/CIP4 homology BAR (F-BAR) and inverse BAR (I-BAR) proteins. The N-BAR and F-BAR domains induce positive membrane curvature through concave lipid-binding interfaces and trigger cell membrane invagination. However, I-BAR domains interact with shallow negatively curved membranes through convex lipid-binding interfaces, leading to cell membrane protrusion [13,14]. Pioneering work in the corn smut fungus,

*Ustilago maydis*, revealed the importance of endocytosis for the pathogenic development and virulence of filamentous fungi by showing impaired early pathogenicity and germination in endocytic mutants [15]. The F-BAR protein, Cdc15, is involved in cytokinetic ring and septa formation in *U. maydis* [16]. Recent studies in *Magnaporthe oryzae* revealed the importance of N-BAR domain-containing proteins in the growth and virulence of filamentous fungi [17]. Further, the I-BAR protein, Rvs167, was found to be involved in the extension of the rigid penetration peg required during *M. oryzae* invasion [18]. The F-BAR-like protein Bzz1p in yeast and Cip4 in *Drosophila* act during the early stages of endocytosis in the formation of actin patches [19,20]. They trigger actin polymerization via the Arp2/3 complex [21]. There has been an intense study on the structural and biochemical properties of BAR domain proteins that contribute to their mode of action [11,13]. However, there has been limited functional characterization of BAR superfamily proteins in phytopathogenic fungi.

*Ascochyta rabiei* (Pass.) Labr. (teleomorph *Didymella rabiei*), a causal agent of Ascochyta blight (AB) disease in chickpea (*Cicer arietinum* L.), is one of the most devastating necrotrophic phytopathogens. *A. rabiei* infects the above-ground parts of this legume plant and greatly reduces the yield of the crop [22]. The fungal hyphae aggregate in the cortical cells of the chickpea plant and differentiate into asexual spores called pycnidia [23,24]. The genome of *A. rabiei* has been sequenced and analyzed to identify pathogenic determinants [25]. *A. rabiei* has emerged as an interesting model system for elucidating the cell biology, especially the endocytic machinery, during polar growth and pathogenesis in necrotrophic phytopathogenic fungi.

In this study, a F-BAR domain-containing protein, ArF-BAR, was identified and characterized in *A. rabiei*. A loss-of-function mutation in *ArF-BAR* gene cause delayed septa formation, severely compromised fungal virulence on chickpea and secretion of Ar93, which is a candidate effector of *A. rabiei* having a secretory signal peptide sequence. The results showed that the F-BAR domain of the ArF-BAR protein binds to and deforms synthetic liposomes and generates membrane tubules. *ArF-BAR* transcripts were induced in response to oxidative stress and infection, and the ArF-BAR protein localized to endocytic vesicles within fungal hyphae. It was also found that *ArF-BAR* expression was regulated by a stress inducible transcription factor, ArCRZ1. Our data suggested that ArF-BAR-dependent actin cytoskeleton dynamics combined with endocytosis and effector secretion is crucial for *A. rabiei* virulence.

## Results

### *ArF-BAR* expression is induced in response to oxidative stress and infection

Transcriptome analysis of *A. rabiei* under oxidative stress has provided a greater understanding of the survival strategies used by necrotrophic fungi against host-generated oxidative stress [26]. The study by Singh et al., [26] revealed that 70 unigenes were upregulated under oxidative stress conditions. Of these 70 unigenes, an expressed sequence tag (EST) matched with gene model "ST47_g8005" of the sequenced *A. rabiei* genome [25] showed early upregulation against oxidative stress. The deduced amino acid sequence of this gene revealed a protein with four distinct domains: An N-terminal F-BAR domain (amino acids 7–253), a unique C1 domain (amino acids 407–459) showing similarity to the C1 domain of Protein Kinase C1 (PKC1), and two consecutive C-terminal SH3 domains (amino acids 572–651 and 705–760) (Fig 1A). Henceforth, this protein has been named as ArF-BAR. Phylogenetic analysis of selected BAR proteins from pathogenic fungi and other eukaryotes revealed that ArF-BAR shared sequence identity with proteins of many closely related fungal pathogens (S1 Fig). ArF-BAR was found to share approximately 33% and 41% sequence identity with

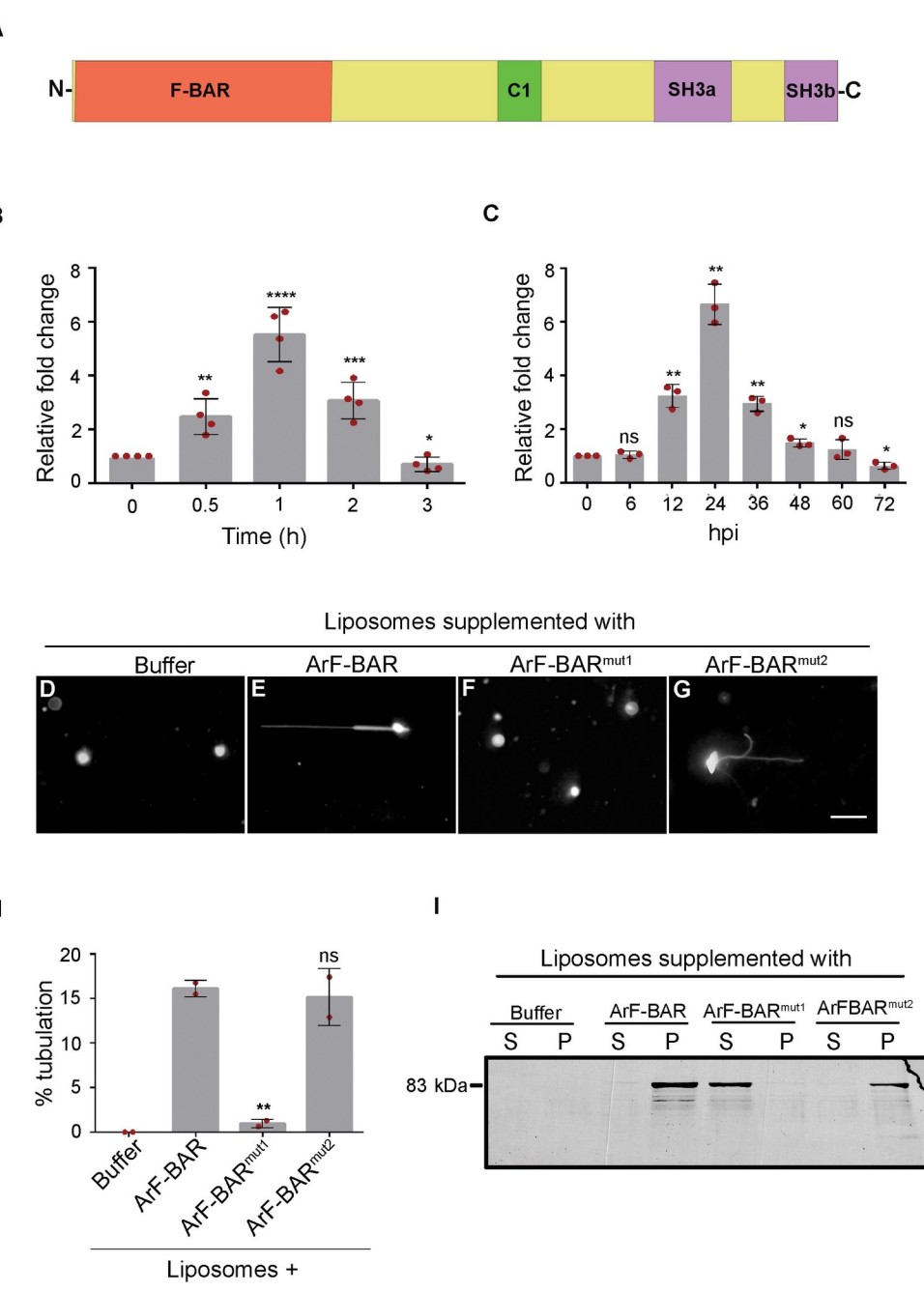

**Fig 1. ArF-BAR is a bonafide stress induced membrane tubulating protein.** Schematic representation of domain organization of ArF-BAR protein. It has a F-BAR domain (amino acids 7–253), a C1 domain of Protein Kinase C1, PKC1 (amino acids 407–456) and two consecutive SH3 domains (SH3a; amino acids 572–651 and SH3b; amino acids 705–760). (B) Relative fold change in the *ArF-BAR* transcript level of 5 days axenically grown potato dextrose broth mycelium culture of *Ascochyta rabiei* post 250 µM menadione treatment. The treated mycelia samples were collected every 0.5 h and the expression was analysed with qRT-PCR. (C) Relative fold change in the transcript level of *ArF-BAR* post *A. rabiei* conidial suspension inoculation. The conidia inoculated chickpea aerial tissue was collected up to 72 hpi and expression was analysed by qRT-PCR. (D-G) Membrane deforming activity of ArF-BAR. The bar represents 20 µm. (D) The liposomes remain spherical in presence of protein resuspension buffer, acting as control. (E, G) Tubulation of synthetic liposomes after addition of the recombinant ArF-BAR and ArF-BAR^mut2 proteins. (F) Liposome tubulation activity is insignificant in ArF-BAR^mut1. (H) The bar chart shows the percentage of the tubule formed by liposomes in presence of native and mutated recombinant proteins. Out of 150 liposomes, the percentage of the tubule formation was counted in each set of experiment. (I) Liposome co-sedimentation assay, where the ArF-BAR and ArF-BAR^mut2 recombinant protein bound with lipid membranes (liposome) were found in pellet, while

membrane binding activity is lost in ArF-BAR$^{mut1}$ protein. Statistical analysis was performed using Student's t-test one tailed compared with its control. Significant differences are indicated as *$p$ = 0.03; **$p$ = 0.003; ***$p$ = 0.0005; ****$p$ = 0.0001; ns = non-significant. Data is the mean of three independent biological replicates with error bars ± representing standard deviation. Red dots represent the average value of all technical replicate in each biological replicate used for the quantitative analysis.

*Saccharomyces cerevisae* BZZ1p and *Drosophila melanogaster* Cdc42-Interacting Protein 4 (CIP4) proteins, respectively (S2 Fig).

To validate the results of the transcript profiling, quantitative real-time PCR (qRT-PCR) was performed using primers specific to the F-BAR domain encoding region. The qRT-PCR experiments revealed a higher expression level of the *ArF-BAR* transcript in *A. rabiei* mycelia after a 1 h treatment with menadione, which is an oxidative stress generator (Fig 1B). To directly assess *ArF-BAR* transcript induction during pathogen proliferation on a susceptible chickpea variety (PUSA-362), the time course of *ArF-BAR* transcript expression was measured using qRT-PCR after inoculating plants with *A. rabiei* conidial suspension. Consistent with the previous observations [26], a significant increase in the *ArF-BAR* transcript level was found following infection. The maximum transcript level was at 24 hours post-inoculation (hpi; Fig 1C), which is within the critical time period for spores to germinate on the host surface and early germinated conidial hyphae starts penetrating chickpea tissue [24,27].

## ArF-BAR is a membrane tubulating protein

To understand whether ArF-BAR has a functional F-BAR domain, synthetic liposomes containing Rhodamine B-conjugated PE were incubated with purified His-tagged recombinant ArF-BAR protein (S3A Fig). Narrow tubules were formed after 10 min of liposomes incubation with purified recombinant ArF-BAR protein (Fig 1D and 1E). The spherical liposomes transformed into an intense network of narrow tubules within 30 min of incubation with ArF-BAR (S3B Fig). This indicated that ArF-BAR protein has liposome tubulation activity. Additionally, the sequence alignment of ArF-BAR with that of other characterized F-BAR domain-containing proteins, revealed the presence of positively charged conserved residues (S2 Fig). In previous studies, the positively charged residues of FBP17 and Syndapin have been shown to interact with negatively charged phospholipids to induce membrane curvature [28,29]. Since ArF-BAR has conserved positively charged amino acids at positions 57, 58, 131 and 132 (S2 Fig), the role of these residues in membrane deformation was evaluated. This evaluation was performed by replacing the residues with glutamate (K57E, K58E, R131E and K132E) via site-directed mutagenesis, and the mutant version was referred to as ArF-BAR$^{mut1}$ (S3C Fig). As expected, it was found that substituting the conserved lysine/arginine with glutamate abolished the tubulation activity of the F-BAR domain (Fig 1F). Protein kinase C1 (PKC1) proteins are known to interact with diacylglycerol (DAG), which has a role in membrane interactions. To abolish the involvement of the unique C1 domain of ArF-BAR in membrane deformation, residues at positions W428 and L430 were replaced with glycine [30], and the mutant form was referred to as ArF-BAR$^{mut2}$ (S3D Fig). Further, this ArF-BAR$^{mut2}$ protein was used to assess the *in vitro* liposome tubulation activity and a similar tubulation activity to that of the wild-type ArF-BAR protein was found (Fig 1G). Thus, positively charged amino acids of F-BAR domain are involved in ArF-BAR mediated membrane tubulation (Fig 1H). To test whether the tubulation activity of ArF-BAR correlated with its lipid binding ability, a liposome co-sedimentation assay was used (Fig 1I). Compared to the native ArF-BAR protein, ArF-BAR$^{mut1}$ showed reduced lipid sedimentation efficiency. In contrast, the sedimentation was unaffected with ArF-BAR$^{mut2}$. This indicated the importance of the direct interaction of

the BAR domain with the lipids in liposome tubulation (Fig 1H and 1I). Together, these results suggest that F-BAR domain of ArF-BAR plays a major role in interacting with the lipid membranes for binding and deformation.

## ArF-BAR is required for *A. rabiei* virulence on chickpea

To elucidate the biological importance of *ArF-BAR* in fungal virulence, an *A. rabiei* knockout mutant (Δ*arf-bar*) was generated. The *ArF-BAR* gene was targeted for deletion using a homologous recombination approach. The results confirmed that the open reading frame (ORF) of *ArF-BAR* had been successfully replaced with a single copy of the hygromycin resistance gene (*Hph*; S4A and S4B Fig). Simultaneously, Δ*arf-bar* mutant strain was complemented with a T-DNA cassette that expressed *ArF-BAR* under its own promoter (Δ*arf-bar/ArF-BAR*; S5A and S4B Figs).

To assess the effect of *ArF-BAR* deletion on vegetative growth of *A. rabiei*; wild-type (WT), Δ*arf-bar*, and knockout complemented (Δ*arf-bar/ArF-BAR*) strains were grown on potato dextrose agar (PDA) plates. The knockout mutants showed reduced radial growth compared to *A. rabiei* wild-type (WT) and this radial growth was restored in the Δ*arf-bar/ArF-BAR* (Fig 2A). To determine the Ascochyta blight (AB) disease development and virulence of the fungal strains, conidial suspensions of each strain were spray inoculated on susceptible chickpea variety plants individually. The typical AB disease symptoms, described previously [23,24], were observed on chickpea plants challenged with the *A. rabiei* wild-type (WT) and Δ*arf-bar/ArF-BAR*, but not on plants challenged with Δ*arf-bar* (Fig 2B). The degree of pathogenicity was quantified according to lesion number and lesion size, which were compared among the *A. rabiei* wild-type (WT) and mutants at 144 hpi. The lesion number per plant in Δ*arf-bar* was significantly lower than in the *A. rabiei* wild-type (WT) (Fig 2C). The mean lesion size was also much lower in Δ*arf-bar* than in the *A. rabiei* wild-type (WT) (Fig 2D). However, for the Δ*arf-bar/ArF-BAR* complementation mutant, the disease symptoms were comparable to those of the *A. rabiei* wild-type WT. Overall, the disruption of the *ArF-BAR* gene significantly impaired the pathogenicity of *A. rabiei*.

To ascertain the role of F-BAR and unique C1 domains in fungal virulence, the two domains were independently inactivated by site-directed mutagenesis as described earlier. The Δ*arf-bar* knockouts were complemented with residues mutated in F-BAR domain (Δ*arf-bar/ArF-BAR*$^{mut1}$; S5B and S4B Figs) and C1 domain (Δ*arf-bar/ArF-BAR*$^{mut2}$; S5C and S4B Figs). In three independent biological replicates, each replicate having at least 20 plants, the WT strain produced 5.6 times greater number of disease lesions per plant in comparison those produced by Δ*arf-bar* mutant strain (WT = 14.99 ± 0.84; Δ*arf-bar* = 2.66 ± 0.19; $p$ = 0.000536; Fig 2C). As compared to the WT, the virulence of Δ*arf-bar/ArF-BAR*$^{mut1}$ strain was very low (disease lesions/plant = 3.02 ± 0.74; mean lesion size = 1.33 mm ± 0.06). Meanwhile, in the case of Δ*arf-bar/ArF-BAR*$^{mut2}$ strain the number of lesions per plant was significantly lower (8.833 ± 0.04; $p$ = 0.000127) than in the WT, but there was no significant difference in the size of the lesions (WT = 3.67 mm ± 0.46; Δ*arf-bar/ArF-BAR*$^{mut2}$ = 2.97 mm ± 0.15; Fig 2D). The severity of the disease symptoms increased in the plants after 10 days post infection (dpi), but no significant differences were observed for plants challenged with Δ*arf-bar* strain (S6 Fig). Together, these results confirm that *ArF-BAR* is an important virulence determinant and that the conserved positively charged residues of F-BAR domain are required for full-virulence of *A. rabiei* on chickpea.

It was subsequently hypothesized that the reduced virulence in Δ*arf-bar* could have resulted from at least two factors: a) knockout mutants may have had a compromised ability to penetrate host tissue, or b) the reduced virulence of the mutant was a consequence of a reduction in

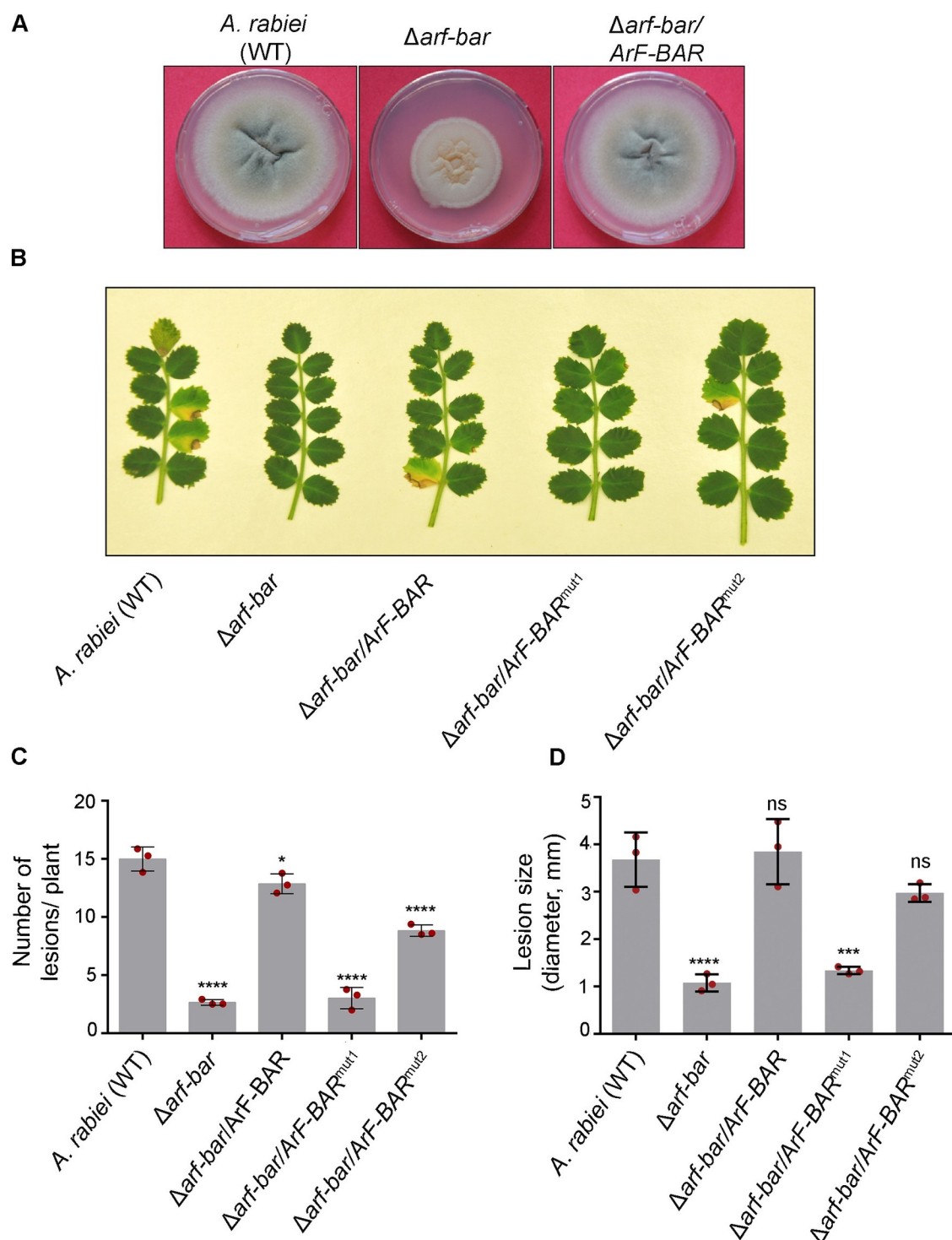

**Fig 2. *ArF-BAR* is required for full virulence of *A. rabiei* in chickpea.** (A) To examine radial vegetative growth on PDA plates, *A. rabiei* (WT), Δ*arf-bar* and Δ*arf-bar/ArF-BAR* mutant strains were grown for 7 days at 22˚C. (B) Representative image of Ascochyta blight (AB) disease symptoms on susceptible chickpea plants at 6 dpi, obtained after inoculation with conidial suspension of *A. rabiei* (WT) and mutant strains. Decreased AB symptoms in Δ*arf-bar* and Δ*arf-bar/ArF-BAR*[mut1] shows reduced virulence. (C) Statistical values of the number of lesions per plant after conidial inoculation of chickpea with *A. rabiei* (WT), mutant and mutant complemented strains. The graph represents the mean and standard deviation of three independent biological replicates, counting at least 20 plants in each replicate. (The bars in graph correspond to 14.99 ± 0.84 in WT; 2.66 ± 0.19 in Δ*arf-bar*; 12.86 ± 0.69 in Δ*arf-bar/ArF-BAR*; 3.02 ± 0.74 in Δ*arf-bar/ArF-BAR*[mut1] and 8.833 ± 0.04 in Δ*arf-bar/ArF-BAR*[mut2]). (D) Statistical values of the lesions'

diameter were calculated (The bar in graphs corresponds to; WT = 3.67 mm ± 0.46; Δ*arf-bar* = 1.07 mm ± 0.14; Δ*arf-bar/ArF-BAR* = 3.84 ± 0.54; Δ*arf-bar/ArF-BAR*mut1 = 1.33 ± 0.06 and Δ*arf-bar/ArF-BAR*mut2 = 2.97 ± 0.15). Statistical analysis was performed using ordinary one-way ANOVA compared with its control (****$p < 0.0001$; ***$p < 0.001$; *$p < 0.05$). Red dots represent the average value of all technical replicate in each biological replicate used for the quantitative analysis.

fungal viability or inability to counter the host basal defense. To address these possibilities, hyphal penetration was examined and compared by inoculating chickpea leaves with the WT and Δ*arf-bar* mutant strains. The leaves of the chickpea plants challenged with these strains were subjected to wheat germ agglutinin-Alexa Fluor conjugate (WGA-AF488) staining. At 48 hpi, the infected leaves were stained with WGA-AF488 to enable visualization of the fungus [31]. The infected leaves were optically sectioned using confocal microscopy starting from the surface of the leaves. It was consistently observed that hyphae of the *A. rabiei* wild-type (WT) strain efficiently penetrated into the chickpea leaves up to depths of 20.25 μm ± 3.92 (mean ± SD; n = 6) (S7A and S7C Fig). In stark contrast, the hyphae of the Δ*arf-bar* mutant were unable to penetrate beyond 9.83 μm ± 2.54 ($p = 0.0079$; S7B and S7C Fig). Thus, the depth and efficiency of hyphal penetration by Δ*arf-bar* were significantly impaired.

Further, to investigate the direct involvement of *ArF-BAR* in fungal viability under oxidative stress conditions, radial growth assays were performed for WT, Δ*arf-bar*, Δ*arf-bar/ArF-BAR*, Δ*arf-bar/ArF-BAR*mut1, and Δ*arf-bar/ArF-BAR*mut2 strains. These strains were inoculated on either potato dextrose agar (PDA) or PDA supplemented with menadione (250 μM and 500 μM) or 2 mM $H_2O_2$. The radial growth was analyzed at 10 dpi. The radial growth of Δ*arf-bar* was reduced compared to that of the WT. The mycelial growth was restored in the wild-type *ArF-BAR* complemented (Δ*arf-bar/ArF-BAR*) and *ArF-BAR*mut2 complemented (Δ*arf-bar/ArF-BAR*mut2) strains (Fig 3A and 3B). However, the mycelial growth of complemented Δ*arf-bar/ArF-BAR*mut1 was unable to match that of the WT. Exposure to oxidative stress led to further growth inhibition in Δ*arf-bar* (Fig 3). The phenotype of complemented strain, Δ*arf-bar/ArF-BAR*, under oxidative stress suggest that cellular processes regulated by ArF-BAR are important for oxidative stress tolerance (Fig 3). Further, exposure to oxidative stress led to greater growth inhibition in Δ*arf-bar/ArF-BAR*mut1 than in the wild-type. However, the growth inhibition of Δ*arf-bar/ArF-BAR*mut2 was similar to that of wild-type. The negative effect on growth of fungal strains till 10 dpi was more pronounced for $H_2O_2$ as compared to menadione (Figs 3 and S8). These results strongly support the hypothesis that *ArF-BAR* is required for full virulence of *A. rabiei* wild-type (WT) on susceptible chickpea host and is required for coping with the host basal defense, mainly oxidative stress.

## Absence of *ArF-BAR* perturbs septa formation

Having a potential role in fungal virulence, the next aim was to determine the subcellular localization of ArF-BAR inside *A. rabiei* wild-type (WT) hyphal cells. The ectopic expression of EYFP decorated ArF-BAR chimeric protein was achieved by *Agrobacterium tumefaciens*-mediated transformation (ATMT) of *ArF-BAR:EYFP* T-DNA cassette in Δ*arf-bar* strain [27]. The bioimaging of the enhanced yellow fluorescent protein tagged ArF-BAR showed punctate distribution throughout the cytoplasm of the fungal hyphae. The chimeric protein was mostly concentrated at the growing hyphal tip and at the septa (Fig 4A and 4B). Fungal hyphae transformed with chimeric ArF-BARmut1::EYFP exhibited disrupted localization of these punctate structures and the fluorescence was completely diffused throughout the cytoplasm. However, the distribution of the fluorescent puncta was unaffected in hyphae of strains stably expressing ArF-BARmut2::EYFP (Fig 4A). In this strain, the punctate structures were distributed throughout the cytoplasm and were prominently concentrated at the growing hyphal tip (Fig 4A).

**A**

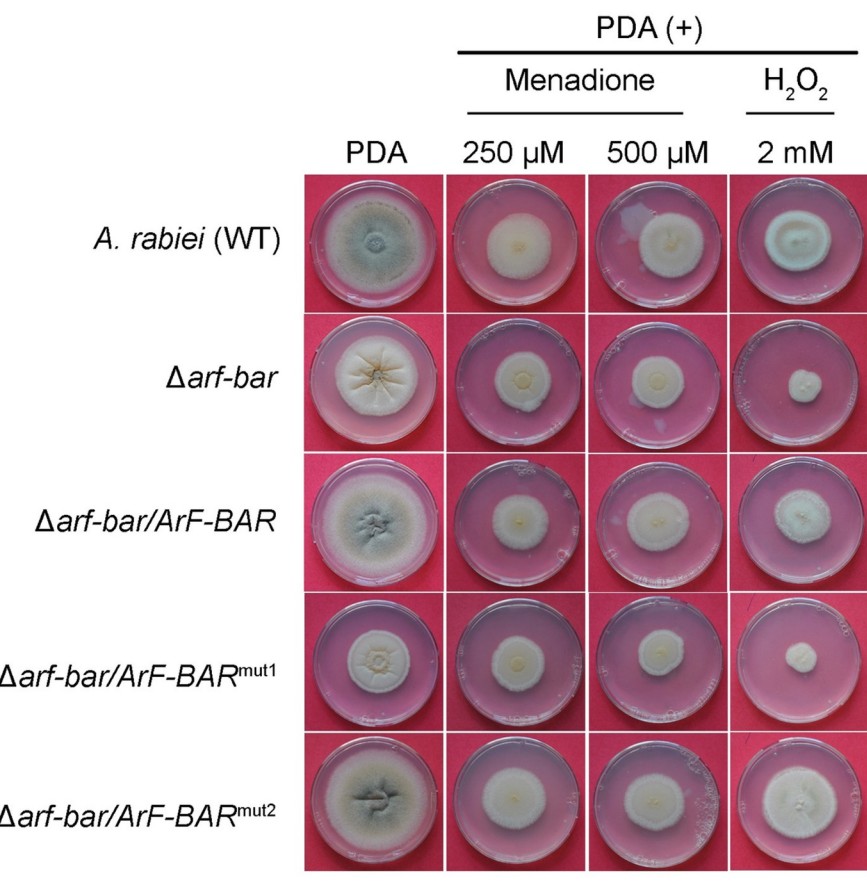

**B**

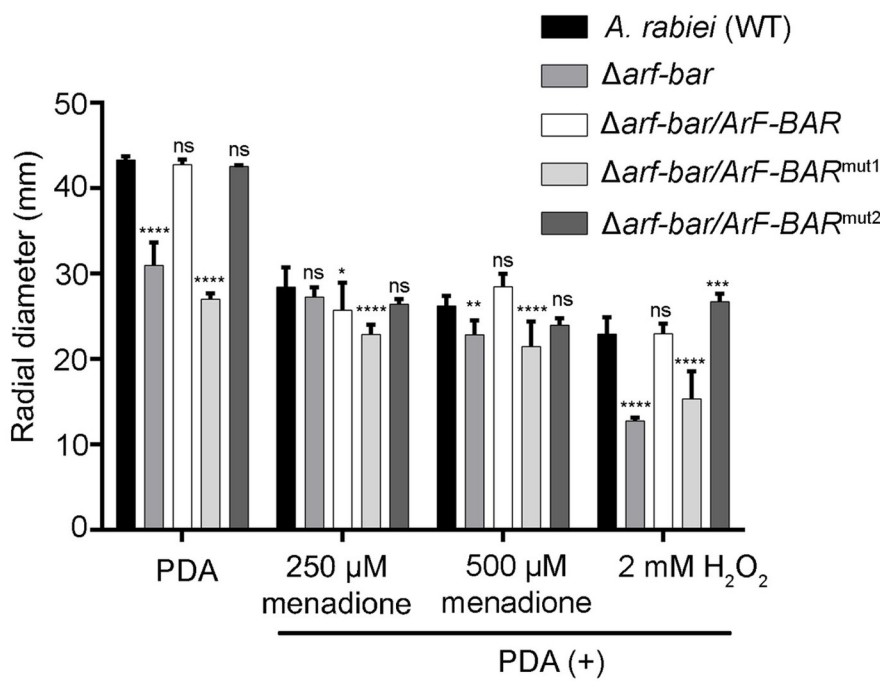

**Fig 3. *ArF-BAR* is required for vegetative growth and oxidative stress tolerance.** (A) Colony morphology and growth assay of *A. rabiei* (WT), Δ*arf-bar* mutant and mutant complemented strains, observed 10 days after incubation at 22˚C. For oxidative stress, the PDA medium was supplemented with 250 μM and 500 μM menadione, and 2mM $H_2O_2$. Strains of Δ*arf-bar* and Δ*arf-bar/ArF-BAR*$^{mut1}$ exhibited enhanced sensitivity towards oxidative stress condition as compared to WT *A. rabiei*. (B) The bar graph represents mean diameter and SD values of fungal strains radial growth in PDA plates and PDA supplemented with menadione or $H_2O_2$. All the PDA growth assays were performed in triplicate. The results were quantified using two-way ANOVA, Tukey's multiple comparisons. $^{****}p < 0.0001$, $^{***}p < 0.001$, $^{**}p < 0.01$, $^{*}p < 0.05$, ns = non-significant.

Further, to validate the spatiotemporal distribution pattern of the ArF-BAR protein during infection, fungal spores expressing ArF-BAR::EYFP were allowed to infect susceptible chickpea stem peel. Microscopic observation of these stem peel infected with fungal hyphae revealed a distribution pattern that was similar to that of the ArF-BAR::EYFP expressing hyphae growing on glass slides; the chimeric protein was predominantly distributed at the hyphal tip and septa (S9 Fig). The localization of ArF-BAR::EYFP towards the membrane of growing hyphae prompted us to check its localization with a Spitzenkörper marker ArSNC1, which is a vesicle-bound v-SNARE protein known for its role in polarized exocytosis in conjugation with other proteins [32–34]. ArSNC1 was identified, from *A. rabiei* genome, based on its high homology with the *M. oryzae* SNC1 (MGG_12614.6) encoded protein. The localization pattern of ArF-BAR::EYFP and ArSNC1::mCherry confirms the apical localization of ArF-BAR (Fig 4C). The *in vitro* germination and growth of WT and Δ*arf-bar* conidia, up to 12 h on hydrophobic surface showed marginal difference (S10 Fig). However, the first cell from the growing hyphal tip in Δ*arf-bar* was observed to be elongated as compared to the WT, which may be due to delayed septation in Δ*arf-bar* as ArF-BAR localizes to the septa. The arrangement of septa within the hyphae of both the WT and the Δ*arf-bar* strain was examined in a number of growing fungal tips. Microscopic analysis of the Δ*arf-bar* mutant using calcofluor white, which precisely stains cell wall components, showed that the hyphae lacked regularly spaced septa (Fig 5B). Interestingly, the distance of the first septum from the growing hyphal tip (polarized end) was significantly greater in the Δ*arf-bar* mutant than in the WT (WT = 23.5 μm ± 3.23 and Δ*arf-bar* = 58.5 μm ± 4.65; mean ± SEM; n = 40 and $p < 0.0000496$; Fig 5C). It showed that Δ*arf-bar* mutant has larger cell length of first and second cells, from the hyphal tip, as compared to *A. rabiei* wild-type (WT) (Fig 5C). Overall, these results reveal that the *ArF-BAR* gene is necessary for appropriate hyphal architecture, which is in turn, important for host penetration and virulence.

## The Δ*arf-bar* mutant impairs secretion of a candidate effector from *A. rabiei*

The delayed proliferation of Δ*arf-bar* strain compared to that of the WT, on susceptible plant surface and less penetration of host tissue (Figs 5A and S7) points towards mutant strain's defect in normal development or suppression of host defense. Therefore, the substantial reduction in virulence of Δ*arf-bar* on chickpea led us to hypothesize that the deletion mutant may be impaired in the ability to secrete effectors compared to *A. rabiei* wild-type (WT). To confirm the likely role of ArF-BAR in effector secretion, a candidate secretory effector Ar93 (Accession number: GW996416) [26] possessing a secretory signal peptide for secretion was ectopically expressed in WT and Δ*arf-bar* backgrounds. The secretion of Ar93 protein fused with EYFP::FLAG to the culture media was examined using concentrated proteins from culture filtrate and cell lysates of WT and Δ*arf-bar*. Western blots analysis of culture filtrate of axenically grown stable fungal transformants (WT and Δ*arf-bar*) showed exclusive secretion of Ar93 effector only in WT transformed strain. However, Ar93 was observed in cell lysates of

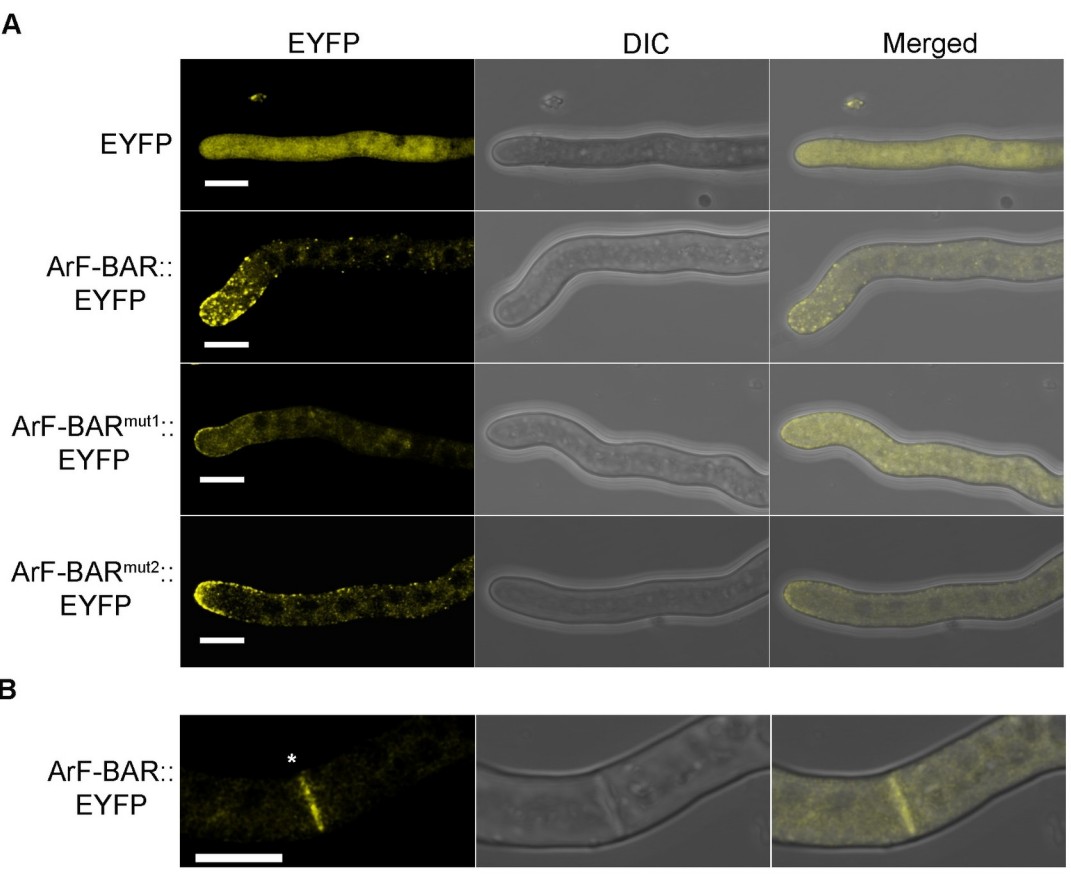

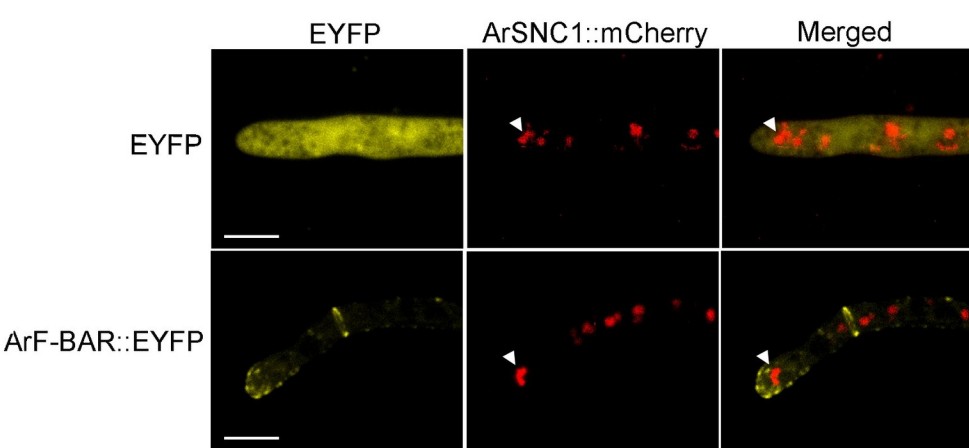

**Fig 4. ArF-BAR primarily decorates the growing hyphal tip.** (A) Ectopically expressed ArF-BAR::EYFP and ArF-BAR$^{mut2}$:: EYFP chimeric proteins are distributed at the hyphal tip. Fungal cells expressing ArF-BAR$^{mut1}$::EYFP had fluorescence scattered throughout the fungal hyphae. The representative images were captured at 12 h post germination of fungal conidia on microscopic coverslip. Scale bar = 5 μm. (B) ArF-BAR::EYFP signal is uniformly distributed at fungal septum, scale bar = 5 μm. Star represents the hyphal septum. (C) The growing hyphal tip localization of ArF-BAR::EYFP was examined along with *A. rabiei* Spitzenkörper marker, ArSNC1::mCherry. Arrowhead represents the localization of ArSNC1. The v-SNARE SNC1 along with other proteins mediate docking and fusion of vesicles with the plasma membrane target site. Micrographs show representative images of three biological replicates with at least 10 images in each replicate.

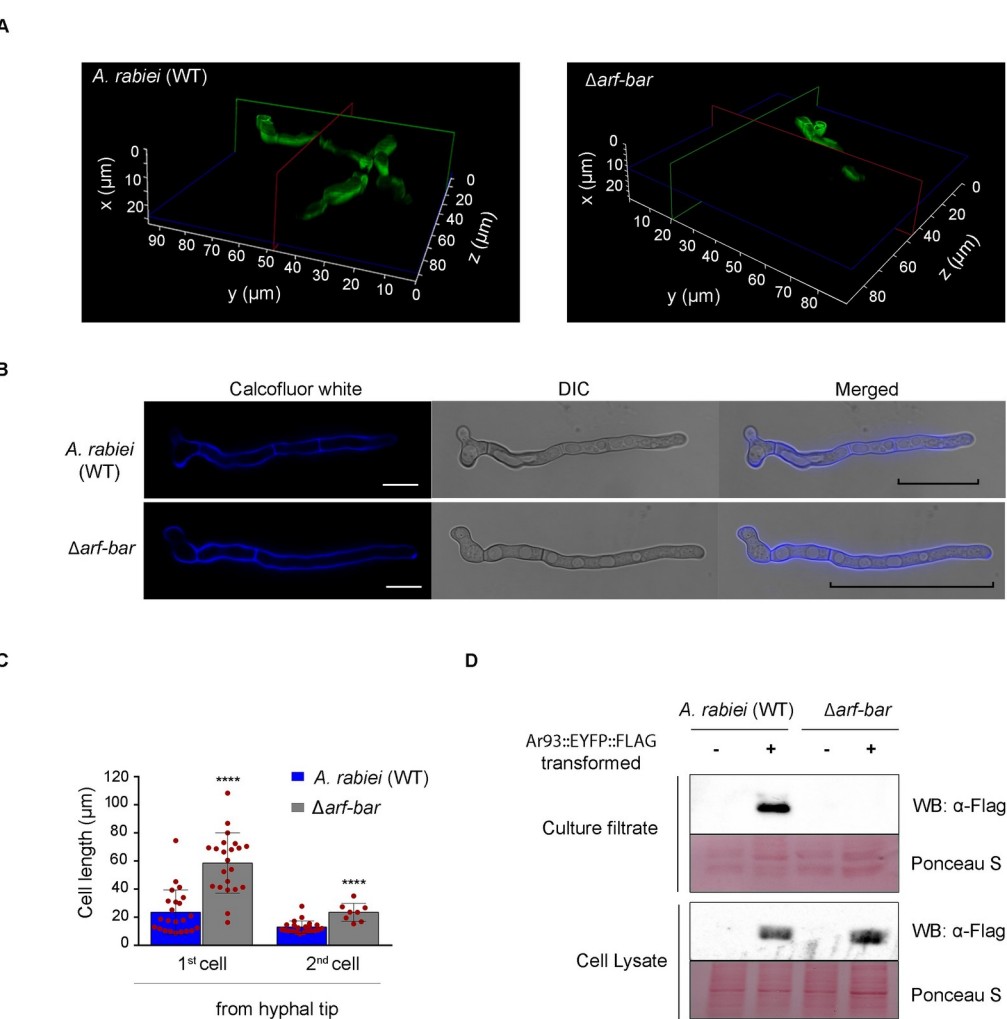

**Fig 5. Deletion of *ArF-BAR* gene perturbs septum formation, host penetration and effector secretion.** (A) 3D structure stacking z-slices together shows the penetration ability of *A. rabiei* (WT) and Δ*arf-bar* on the chickpea host leaves. The Δ*arf-bar* mutant could penetrate on an average *Z* value of 10 μm as compared to WT value of 20 μm. (B) Calcofluor white stained *A. rabiei* (WT) and Δ*arf-bar* hyphae after 12 h post germination. Increase in the distance of first septa from the polarised end was found in Δ*arf-bar*. Black line marks the position of first septa from growing hyphal tip in both WT and Δ*arf-bar*. Scale bar = 10 μm. (C) Bars in graph represent the mean values with SEM, of first cell and second cell length (in μm) from the growing tip end. Significance of the difference in length of cells was calculated using one-tailed paired t-test, (****$p$ = 0.001). Red dots represent the value of each biological replicate used for the quantitative analysis. (D) Secretion of an ectopically expressed candidate secretory effector possessing a classical signal peptide, Ar93::EYFP::Flag (43 kDa) in WT and Δ*arf-bar* mutant. The Δ*arf-bar* strain of *A. rabiei* is impaired in effector secretion in contrast to WT strain. The absence of signal in the western blot in Δ*arf-bar* mutant represents absence of secreted Ar93 effector in culture filtrate, although the chimeric Ar93 is present in the cell lysate of both strains. The minus (–) sign represents Ar93 non-transformed and plus (+) represents Ar93:EYFP:FLAG transformed *A. rabiei* strains. Ponceau S stained blots represent protein loading control.

both the WT and Δ*arf-bar* suggesting Ar93 protein expressed in both strains (Fig 5D). Thus, we conclude that Δ*arf-bar* mutant is impaired in secretion of Ar93 or similar effectors possessing classical secretory signal peptide.

## The Δ*arf-bar* mutant has impaired endocytosis

Since F-BAR domain proteins are known to dimerize and form a canonical banana-shaped fold [35], the dimerization of ArF-BAR was confirmed using a spilt-ubiquitin based yeast two-

hybrid (Y2H) system. The growth of yeast cells on selective media and expression of β-gal reporter gene suggested that ArF-BAR protein dimerize. This evidence further indicates the evolutionarily conserved nature of the BAR proteins (S11 Fig). To gain insight into the role played by ArF-BAR in endocytosis, an amphiphilic styryl type endocytic tracer dye, N-(3-triethylammoniumpropyl)-4-(p-diethyl-aminophenyl-hexatrienyl) pyridiniumdibromide (FM4-64), was used [36]. Microscopic analysis of FM4-64 stained WT and Δ*arf-bar* hyphae revealed internalization of the dye in WT but not in mutant cells. The results suggested rapid internalization of dye in WT hyphae, compared to Δ*arf-bar* strain where no such internalization observed even after 10–15 min of FM4-64 incubation (Fig 6A). To validate these findings, the mean fluorescence intensity area was determined in biological replicates (details in subsection 'quantification and statistical analysis' of Materials and Methods). The quantitative analysis of FM4-64 data suggested that the endocytic activity from the plasma membrane of *A. rabiei* cells was reduced by 49.31% in Δ*arf-bar* mutant strain (n = 12, *p* = 0.0002; Fig 6A). The result provides evidence that *ArF-BAR* regulates endocytosis in *A. rabiei*.

Central components of the endocytic pathway are the EEs, where the small GTPase Rab5 plays a major regulatory role in their biogenesis [37]. *ArRab5*, an orthologue of *Rab5*, was identified in the *A. rabiei* genome using *Rab5* of *M. oryzae* and *U. maydis* as query at NCBI. The relationship between ArF-BAR-associated endocytosis and ArRab5-associated early endosomes was determined using a double-labeling experiment. The T-DNA constructs having *ArF-BAR* tagged with *mCherry* and *ArRab5* tagged with *EGFP* were sequentially transformed into the WT *A. rabiei* strain. The coalescence of fluorescence showed perfect positive correlation, for the co-localization, between ArRab5 and ArF-BAR chimeric proteins (n = 22; Fig 6B).

Early endosomes mature to late endosomes followed by the replacement of Rab5 by Rab7 [38]. Thus, we aimed to determine whether the punctate distribution of ArF-BAR was associated with all endocytic vesicles or specifically to the EEs. In this context, similar to *Rab5* orthologue, an orthologue of *Rab7* was identified in *A. rabiei* and was tagged with EYFP (*ArRab7:EYFP*). A similar double-labeling experiment was performed that showed separate intensity and localization of ArF-BAR::mCherry and ArRab7::EYFP chimeric proteins (Fig 6C). In summary, these results show association of ArF-BAR with early endocytic vesicle protein complexes and defect in FM4-64 endocytosis suggest a role in endocytosis.

## ArF-BAR modulates the actin cytoskeleton

The presence of SH3 (SRC homology 3) domains in F-BAR proteins is well documented for their relationship with the actin cytoskeleton via interactions with the Arp2/3 complex activator Wiskott-Aldrich syndrome protein (WASp) [39]. Since, ArF-BAR in *A. rabiei*, contains two consecutive SH3 domain at its C-terminus, we initially hypothesized the existence of interactions of ArF-BAR with ArActin. Using Y2H system, it was shown that ArF-BAR protein does not interact with actin (Fig 7A), rather it physically interacts with WASp through its SH3 domains (Amino acids 572–760; Figs 7B and S12A and S12B). Further, to check the role of ArF-BAR in actin polymerization, a well-established *in vitro* actin polymerization assay was performed. The kinetics of actin polymerization was monitored by increase in the fluorescence of pyrene-labeled actin. The effect of purified ArF-BAR protein on actin nucleation (actin, Arp2/3 and WASp) was tested using a minimal set of components for all reactions. Interestingly, the addition of purified recombinant protein led to an increase in the actin polymerization rate (Fig 7C). By increasing the concentration of purified protein, a significant gradual activation in the rate of actin polymerization was observed (Fig 7C). These results strongly suggest that ArF-BAR plays an active role in WASp-dependent actin polymerization.

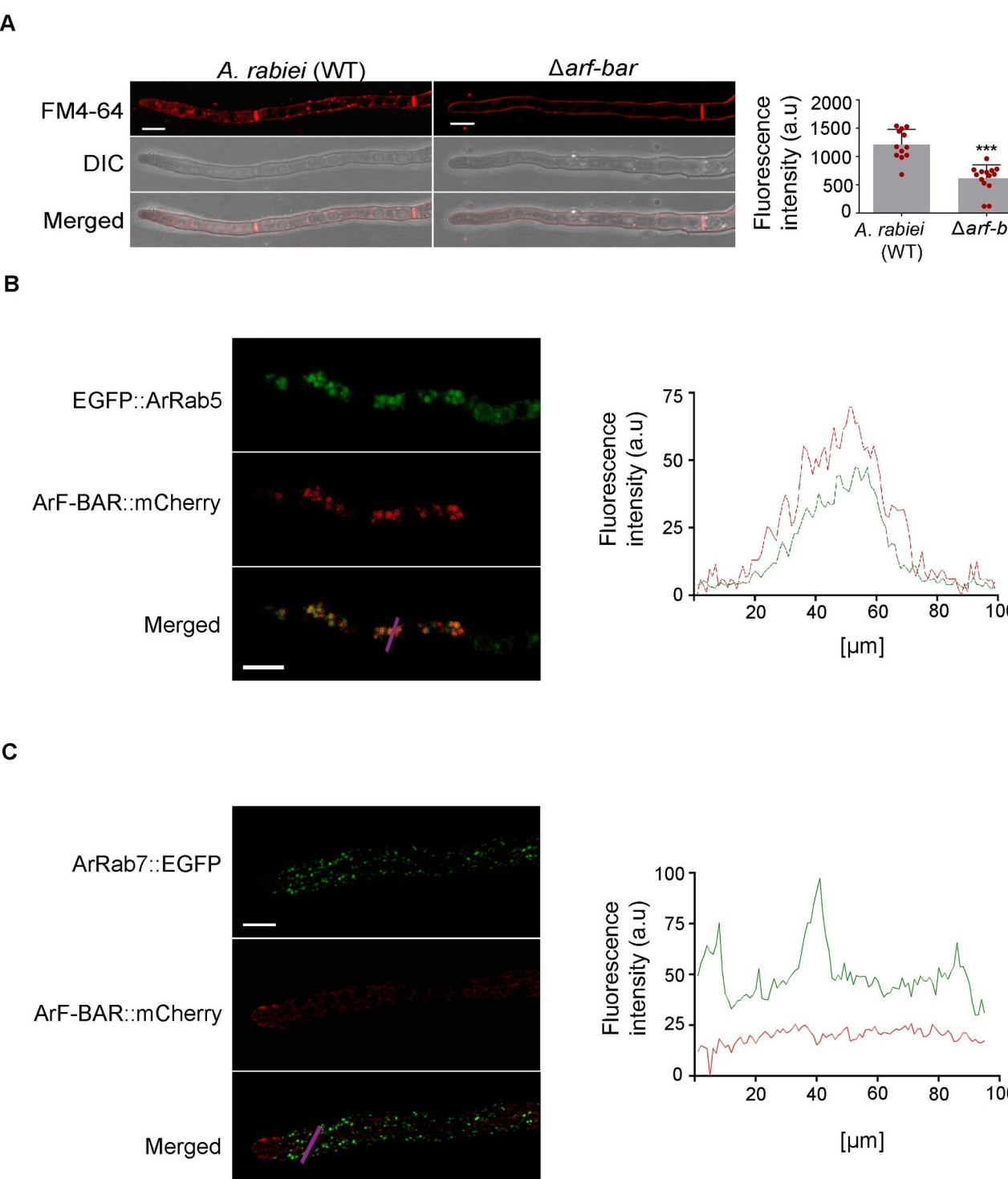

**Fig 6. ArF-BAR facilitate endocytosis.** (A) Confocal images of *A. rabiei* (WT) and Δ*arf-bar* hyphae after 10 min incubation with FM4–64 to acquire internalization capacity. Scale bar = 5 μm. Right panel represents the bar graph of FM4-64 fluorescence intensity per unit area mean value (a. u = arbitrary unit) in WT *A. rabiei* (1210.78 ± 256.18) and Δ*arf-bar* mutant (604.092 ± 246.62) strains hyphae after internalization of the dye. The significance of mean fluorescence intensities variation was statistically examined using one-tailed paired t-test, \*\*\**p* = 0.0002. Red dots represent the value of each replicate. (B) Confocal images showing co-localization of ArF-BAR::mCherry with EGFP::ArRab5 (n = 22), scale bar = 10 μm. Right panel shows the positive correlation between the fluorescence intensity of ArF-BAR (Red) with ArRAB5 (Green) across the magenta line in merged panel. (C) ArF-BAR does not localizes with the late endosome marker ArRab7::EYFP. The representative image is the maximum intensity projection of all *z*-stack

images with 1 μm step size. Images were acquired after 12 h post-germination of fungal conidia expressing ArF-BAR::mCherry and ArRab7::EYFP. Scale bar = 5 μm. The images were acquired from five biological replicates with each having five technical replicates.

Appropriate organization of actin is required for vesicular dynamics, organelle movement and cytokinesis. Actin microfilaments or F-actin are organized into higher order structures comprising of actin patches, cables and rings that serve as the track for long distance transport [40]. To elucidate the relative importance of ArF-BAR in actin organization, actin dynamics was compared in the WT and Δ*arf-bar* mutants. Here, we took advantage of Lifeact, an actin binding peptide fused with a fluorescence protein. Lifeact has been successfully employed for *in vivo* visualization of actin filaments and dynamics in a variety of organisms including fungi and plants [41]. In this study, a *Lifeact*:*mCherry* fusion construct was generated and transformed into both the WT and Δ*arf-bar* mutant to visualize cytoplasmic actin dynamics in fungal hyphae. Confocal microscopy revealed discrete actin patches and cables in the WT hyphae, while actin patches were rarely visible in the Δ*arf-bar* mutant (WT = 92 ± 19.86 and Δ*arf-bar* = 16.3 ± 4.49; *p* = 0.0272; Fig 7D and 7E). Moreover, actin cables were dramatically disorganized in the Δ*arf-bar* mutant. Although ArF-BAR does not directly interact with actin, it regulates actin polymerization and actin cytoskeleton through its association with WASp in growing fungal hyphae.

## ArCRZ1 is a potential transcriptional regulator of *ArF-BAR* expression

Thus far, the findings of the present study have revealed the importance of *ArF-BAR* in the regulation of endocytic pathways, which is crucial for the pathogenesis of *A. rabiei*. However, the transcriptional regulators associated with the expression of endocytic pathway genes in filamentous pathogenic fungi are largely unknown. Hence, the transcriptional mechanism associated with the regulation of *ArF-BAR* expression during host infection was analyzed. The known core-binding motifs for fungal TFs were identified in the 727 bp 5′ upstream regulatory genomic sequence of *ArF-BAR* gene by YEASTRACT online tool [42]. It revealed seven different putative TF binding sites in the 5′ upstream regulatory region between *ArF-BAR* and ST47_g8004 genes (S1 Table). Among these, three core-binding sites (GAGCCC) were found for calcineurin-responsive zinc finger transcription factor 1 (CRZ1), positioned at -106, -136, and -254 upstream to the *ArF-BAR* translational start site ATG (Fig 8A).

CRZ1 is an evolutionarily conserved TF from yeast to mammals. CRZ1 was chosen for analysis because it regulates the expression of various genes involved in stress tolerance [43] and is widely known to translocate inside the nucleus with an increase in cytosolic $Ca^{2+}$ ion concentration [44]. *Ascochyta rabiei* gene (*ArCRZ1*; ST47_g3738) encodes a protein that possesses a serine-rich region (SRR), two consecutive calcineurin docking domains (CDD), characterized by PxIxIT motif (PRILPQ and PEINID) and a single $C_2H_2$ zinc finger motif (S13A Fig). To determine the role of ArCRZ1 in the transcriptional regulation of *ArF-BAR*, the binding of ArCRZ1 to the regulatory sequences of *ArF-BAR* was confirmed. This confirmation was performed via an electrophoretic mobility shift assay (EMSA) using recombinant His-tagged ArCRZ1 proteins. Shifting was observed for the DNA fragment possessing the calcineurin-dependent response element (CDRE) in the presence of purified His-ArCRZ1. However, mutations in this CDRE resulted in the abolishment of binding (Fig 8B and 8C).

Subsequently, the sub-cellular localization of ArCRZ1 under $Ca^{2+}$ and oxidative stress conditions was determined. The *A. rabiei* wild-type (WT) was transformed with a translational fusion of *ArCRZ1* with *EYFP* towards C-terminus. After, 0.2 M $CaCl_2$ treatment, confocal microscopy revealed nuclear localization of the ArCRZ1::EYFP signal (Fig 8D). Interestingly, in absence of any treatment, the ArCRZ1::EYFP fusion protein was uniformly distributed

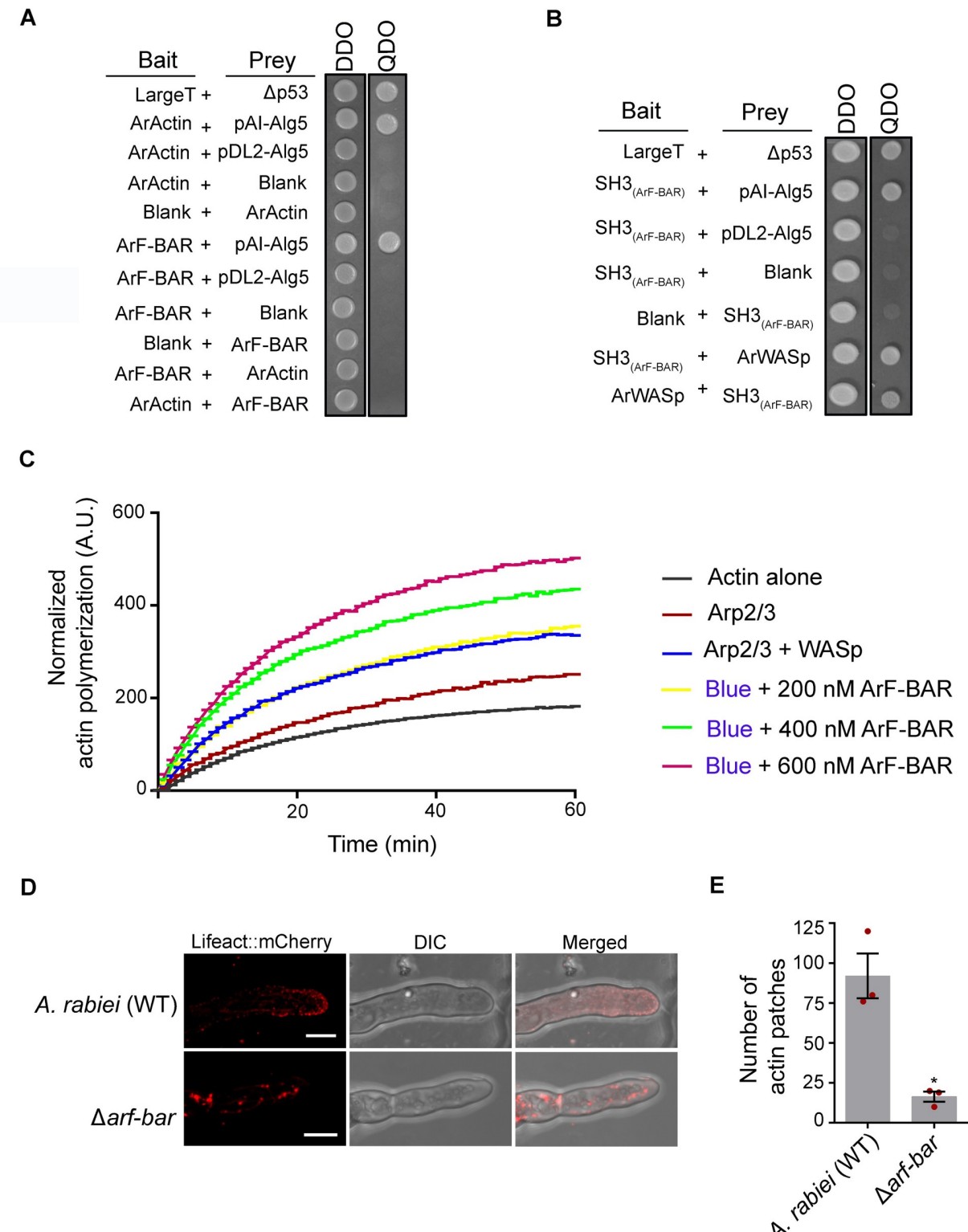

**Fig 7. ArF-BAR modulates the actin cytoskeleton.** (A) Split-ubiquitin based yeast two-hybrid assay to check protein-protein interaction in yeast cytoplasm. The lack of growth for yeast colonies on interaction selective QDO (SD/-L-W-A-H) media as compared to vector selective DDO (SD/-L-W) suggest no interaction between ArF-BAR and ArActin. The LargeT and Δp53 are interaction positive controls while pAI-Alg5 and pDL2-Alg5 are the controls to check bait expression and autoactivation, respectively. (B) ArF-BAR exhibits positive interaction with ArWASp through its SH3 domains [SH3$_{(ArF-BAR)}$; 508–760 amino acids]. The representative images were photographed 48 h after spotting 10 μl yeast cells

suspension. The results were evaluated with three independent biological replicates. (C) Kinetics of WASp and Arp2/3 mediated actin polymerization in absence and presence of ArF-BAR. The kinetics was measured with the change in fluorescence of pyrene-actin. For all *in vitro* actin polymerization assays; 4 μM pyrene labelled actin, 13 nM Arp2/3 and 15 nM WASp were used. All the experiments were performed in three replicates. (D) Fluorescence image of Lifeact::mCherry expressed in fungal hyphae WT and Δ*arf-bar*, where discrete actin patches and cables are visible in WT however actin patches are sparsely visible in Δ*arf-bar*. Scale bar = 5 μm (n = 10). (E) Number of actin patches in hyphae of WT (92 ± 19.86) and Δ*arf-bar* (16.3 ± 4.49) fungal strains, visualized by Lifeact::mCherry fluorescence. Red dots represent the mean numerical value of actin patches in three biological replicates, each having ≥ 8 technical replicates.

within the cytoplasm during normal growth on glass slides (Fig 8D). Since, the nuclear translocation of CRZ1 is mediated by a phosphatase (calcineurin) [45], a chemical genetics approach was used to confirm the relationship between calcineurin and ArCRZ1. A potent calcineurin inhibitor, FK506 (Tacrolimus) was used to inhibit the enzymatic activity of calcineurin [43]. FK506 potentially inhibited nuclear translocation of ArCRZ1::EYFP, which suggests that calcineurin plays a role in the nuclear translocation of ArCRZ1 under stress conditions (Fig 8D). To corroborate and extend these findings during infection, susceptible plants were challenged with fungal conidia expressing ArCRZ1::EYFP and nuclear localization of the chimeric protein was observed (Fig 8E). Overall, these results confirm the evolutionarily conserved regulation of ArCRZ1 by calcineurin under stress conditions.

To uncover the transcriptional regulation of the *ArF-BAR* gene mediated by ArCRZ1, a targeted deletion mutant strain of the *ArCRZ1* gene was generated (Δ*arcrz1*; S13B), followed by complementation of mutant with *ArCRZ1* controlled by native promoter (Δ*arcrz1/ArCRZ1*; S13C and S13D Fig). Interestingly, no significant difference in radial growth diameter was observed between Δ*arcrz1* mutant and *A. rabiei* wild-type (WT) (Fig 9A). The expression pattern of *ArF-BAR* in the WT and Δ*arcrz1* was analyzed using semi-quantitative RT-PCR. The results clearly showed significant reduction in *ArF-BAR* transcription in the Δ*arcrz1* mutant (Fig 9B). Imposing oxidative stress to both the WT and the Δ*arcrz1* mutant via menadione treatment revealed an upregulation of the *ArF-BAR* transcript in the wild-type. However, this upregulation was found to be completely abolished in the Δ*arcrz1* mutant (Fig 9B). Overall, these results substantiate that under oxidative stress conditions, the transcriptional regulation of *ArF-BAR* gene is mediated by the activated ArCRZ1.

## Loss-of-function of *ArCRZ1* compromises virulence similar to Δ*arf-bar*

As *ArF-BAR* gene was transcriptionally regulated by ArCRZ1, it was hypothesized that the pathogenicity phenotypes of Δ*arcrz1* should be similar to those of the Δ*arf-bar* mutants. Consistent with this hypothesis, it was observed that Δ*arcrz1* mutants displayed compromised virulence during chickpea infection bioassay. Results showed that the number of lesions per plant (WT = 15.31 ± 1.44 and Δ*arcrz1* = 2.32 ± 0.80; *p* = 0.0008) and size of lesions (WT = 4.04 mm ± 0.41 and Δ*arcrz1* = 2.03 mm± 0.33; *p* = 0.0044) were lower in the Δ*arcrz1* mutant than in the wild-type *A. rabiei*. The data was obtained using ≥ 60 chickpea plants inoculated with each fungal strain in three biological replicates. This virulence defect was rescued in Δ*arcrz1/ ArCRZ1* complemented strain (Fig 9C–9E). The radial growth patterns of Δ*arcrz1* mutants grown on PDA supplemented with Ca$^{2+}$, menadione and sodium dodecyl sulfate (SDS) were also monitored. Under these tested stress conditions, the growth inhibition was significantly greater for Δ*arcrz1* than for *A. rabiei* wild-type (WT). However, the growth phenotype was restored in the Δ*arcrz1/ArCRZ1* strain, which suggests that *ArCRZ1* plays a crucial role in calcium ion signaling, oxidative stress response and maintaining cell-wall integrity during infection (Fig 9F and 9G). Taken together, infection assay and stress treatments demonstrate that ArCRZ1 is a key regulator of *ArF-BAR*, which further regulates various cellular processes like endocytosis, effector secretion, and promotes fungal survival under host generated stress

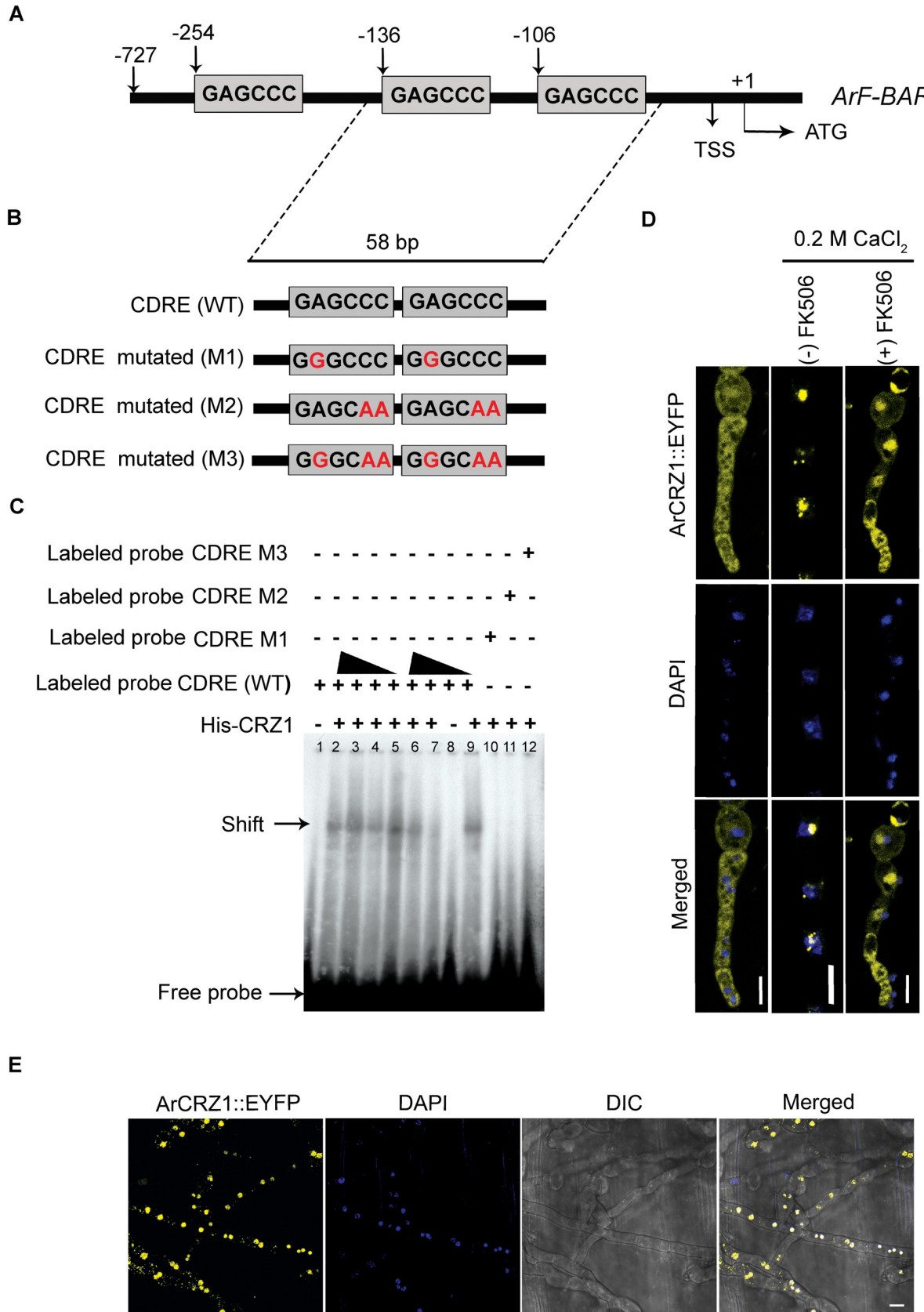

**Fig 8. ArCRZ1 binds *ArF-BAR* promoter and localizes to nucleus under stress conditions.** (A) Schematic representation of ArCRZ1 TF binding sites at the 5′ regulatory region of *ArF-BAR* gene, identified by YEASTRACT online tool in 727 bp sequence. (B) Schematic representation of 58 bp WT and mutated Calcium Dependent Response Element (CDRE) probes used for electrophoretic mobility shift assay (EMSA) of recombinant ArCRZ1. Mutated nucleotides are depicted in red colour. (C) The EMSA blot of recombinant ArCRZ1 with WT or mutated CDRE dsDNA probes. His-purified ArCRZ1 recombinant protein in 500, 600 and 200 ng quantities were used in lane 2, 3 and 4, respectively. The same protein in 800, 200 and 100 ng amounts were used in lane 5, 6 and 7, respectively. The lane 1 and 8 contains only the dsDNA probe with protein. The plus (+) and minus (-) sign representation for labelled probes show the presence and absence of dsDNA probes. The lane 9 has WT CDRE, and lane 10, 11 and 12 have mutated CDRE. This EMSA experiment was performed in triplicate. (D) Confocal images showing the cytosolic distribution of ArCRZ1::EYFP in the absence of stress condition and nuclear localization of ArCRZ1::EYFP in hyphae under stress condition (0.2 M CaCl$_2$). Overnight grown fungal hyphae was exposed to 0.2 M CaCl$_2$ two minutes prior to the microscopy. DAPI fluorescence was simultaneously recorded. Right panel in the confocal images shows the cytosolic distribution of ArCRZ1::EYFP under stress condition (200 mM CaCl$_2$) in the presence of FK506 (5 μg/μl) calcineurin inhibition. Fungal hyphae were exposed to FK506 for 5 min, prior to microscopy. Scale bar = 5 μm. (E) Confocal images showing the nuclear localization of ArCRZ1, during *in planta* fungal growth. Confocal images were acquired 48 hpi of AB susceptible chickpea stem peel, infected with conidia of fungal strain expressing ArCRZ1::EYFP. Scale bar = 5 μm (n = 30).

condition. Thus, ArCRZ1 induces the expression of *ArF-BAR* by directly binding to its *cis*-regulatory region and is critical for *ArF-BAR* dependent virulence.

## Discussion

Dynamic membrane remodeling is essential for maintaining the integrity and identity of cells and cellular compartments [40]. Biological macromolecules, such as BAR superfamily proteins, can sense or induce membrane curvature. They are well-known modulators of transient membrane deformation [39]. Emerging evidence strongly suggests that EEs are crucial for long-distance intracellular communication. Thus, EEs and their roles have broad implications for a wealth of cellular processes such as growth, development, and virulence in filamentous fungi [4,46,13]. The results of the present study further corroborate this conclusion and provide new information regarding signaling and transcriptional control of F-BAR proteins in phytopathogenic fungi. To date, our understanding of the role played by F-BAR proteins in membrane curvature generation and efficient long-range endosome trafficking in fungi during plant-pathogen interactions remains very limited.

To best of our knowledge, this study provides the evidence that an F-BAR domain protein acts as a key mandate for virulence of a phytopathogenic fungi. Based on the evidences provided in the present study, we propose that ArF-BAR contributes to endocytosis, effector secretion, septa formation, and actin dynamics during the *A. rabiei* hyphal growth (Fig 10).

In a first set of evidences, our results indicate the role of membrane remodeling in septa formation and hyphal tip growth. Previous studies, largely conducted in animal models, have highlighted that the F-BAR domain is a membrane-deforming module and is involved in endocytosis [47,29]. The endocytic event is crucial for the uptake of signal cues and nutrition from the host, and aids apical recycling of membrane receptors and proteins. This process thus helps to maintain the overall polarity of the hyphae that is required for fungal growth and virulence [6,43]. The generation of EEs and their trafficking involves microtubule dynamics, actin cytoskeleton rearrangements and most importantly, extensive membrane remodeling [48]. Actin dynamics and microtubule organization have been extensively studied. However, the detailed mechanism underlying the functional regulation of fungal EE biogenesis and EE trafficking at the hyphal tip during plant pathogenesis remains poorly understood.

The present study identifies the role for the ArF-BAR protein as an essential component of endocytosis. ArF-BAR in coordination with Arp2/3-WASp, was found to mediate actin cytoskeleton assembly at the hyphal tip of growing end. The Arp2/3-WASp assembly is a prerequisite for host penetration [33,49]. The early endosomes move on molecular tracks of F-actin network [8,39], which was disorganized in the Δ*arf-bar* mutant, affecting endocytic transport.

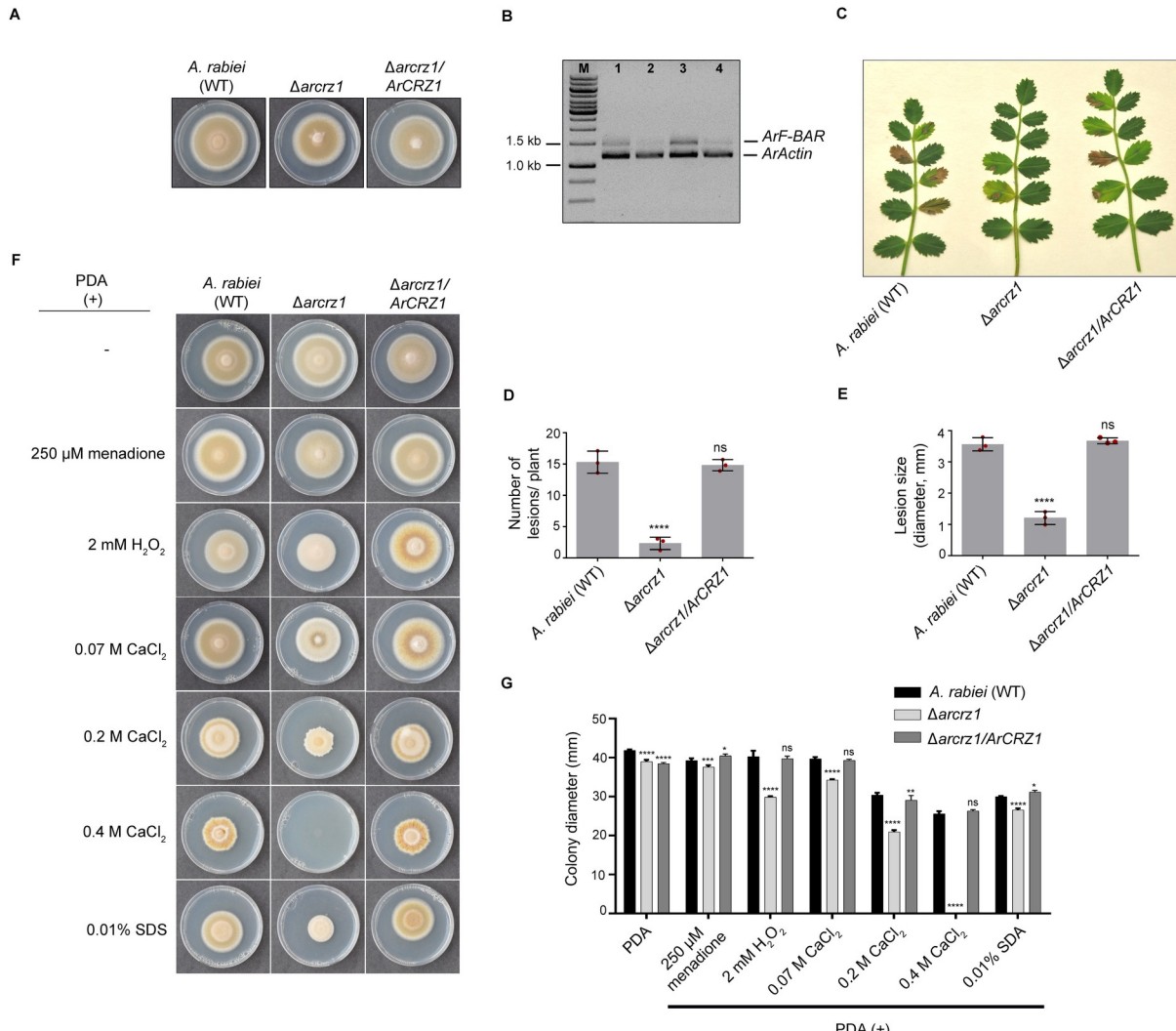

**Fig 9. *ArCRZ1* regulates *ArF-BAR* transcription and *A. rabiei* virulence.** (A) Vegetative growth phenotype of 7 days old *A. rabiei* (WT), Δ*arcrz1* and Δ*arcrz1/ArCRZ1* strains grown on PDA plate. (B) Expression analysis of *ArF-BAR* gene in Δ*arcrz1* through semi-quantitative PCR. Lane 1 and 2 shows *ArF-BAR* expression in mycelia of *A. rabiei* (WT) and Δ*arcrz1* while lane 3 and 4 shows *ArF-BAR* expression in *A. rabiei* (WT) and Δ*arcrz1* after 0.5 h treatment with 250 μM menadione to the axenic culture of mycelia in PDB. (C) Representative image of disease symptoms obtained on Ascochyta blight susceptible chickpea 6 days post inoculation (dpi) of *A. rabiei* (WT), Δ*arcrz1*, and Δ*arcrz1/ArCRZ1* strains. (D) Statistical analysis of the AB disease symptoms on chickpea plants inoculated with three different *A. rabiei* strains. The height of bar in graph shows the mean value of lesions per plant (WT = 15.31 ± 1.44, Δ*arcrz1* = 2.32 ± 0.80, and Δ*arcrz1/ArCRZ1* = 14.82 ± 0.73). (E) The bar height in graph represents the mean value of lesion size, in diameter (WT = 4.04 mm ± 0.41, Δ*arcrz1* = 2.03 mm ± 0.33, and Δ*arcrz1/ArCRZ1* = 3.98 mm ± 0.56). The mean and standard deviation (±) were calculated from three biological replicates, counting at least 20 plants for each replicate. These results were quantified using one-way ANOVA, compared with the control (****$p < 0.0001$). Each red dot represents the mean value of biological replicate. (F) Colony morphology and growth assay of WT *A. rabiei*, Δ*arcrz1* mutant and complemented Δ*arcrz1/ArCRZ1* fungal strains observed under various stress conditions after 10 days incubation at 22°C. PDA medium was separately supplemented with 250 μM menadione, 2 mM $H_2O_2$, 0.07 M, 0.2 M, 0.4 M $CaCl_2$ and 0.01% SDS. The Δ*arcrz1* mutant exhibited enhanced sensitivity towards stress conditions as compared to *A. rabiei* (WT). The mutant strain was highly sensitive towards $CaCl_2$ and failed to grow at 0.4 M $CaCl_2$. (G) The bar graph represents diameter values (mean and SD) of the radial growth for three fungal strains. All the growth assays were performed in triplicates. The results were quantified using two-way ANOVA, Tukey's multiple comparisons. The statically significant differences are shown with ****$p < 0.0001$, ***$p < 0.001$, **$p < 0.01$, *$p < 0.05$, ns = non-significant.

The loss-of-function mutant of the *ArF-BAR* gene leads to the substantial attenuation of virulence compared to WT. This loss in virulence is similar to that observed in the rice blast fungus *M. oryzae* and *U. maydis* where endocytosis is crucial for the recognition of host partners

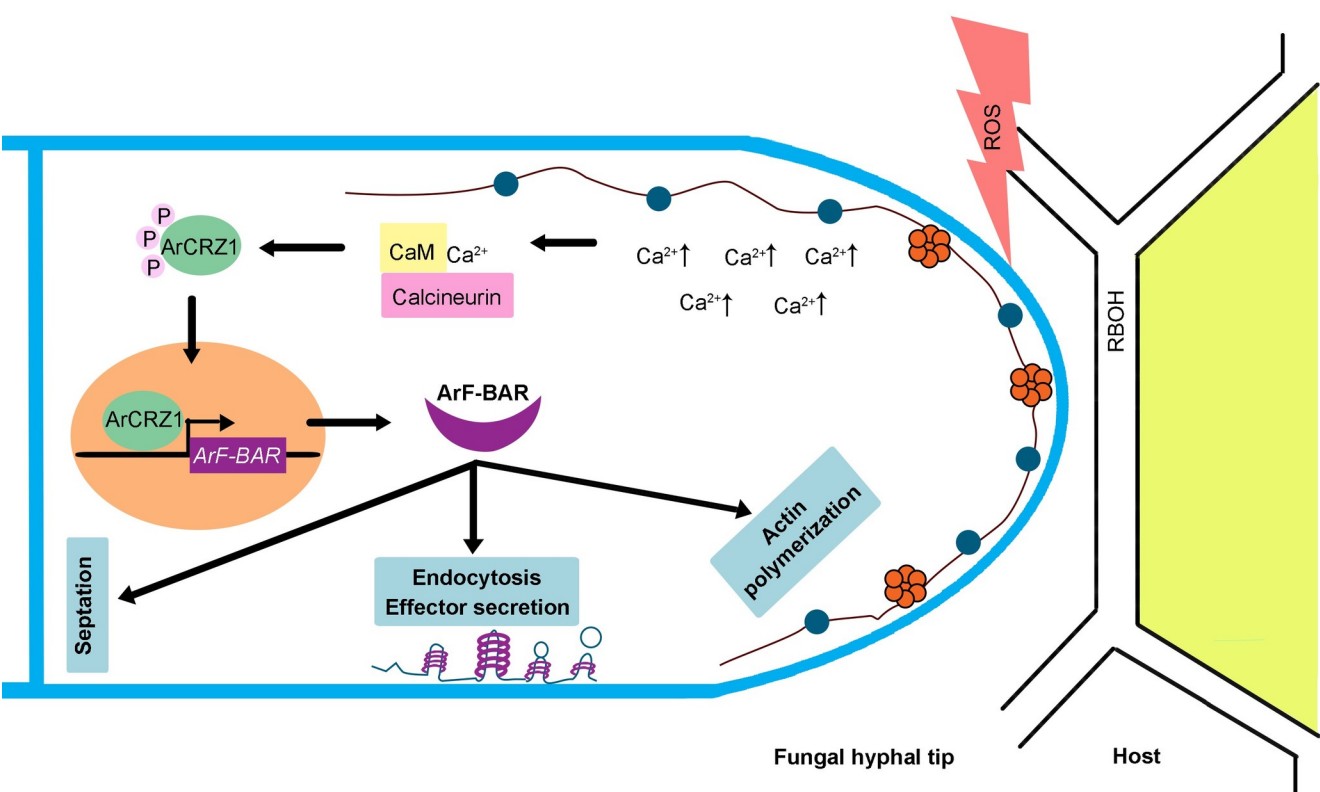

**Fig 10. ArF-BAR regulated functions in *A. rabiei* cells and its regulation by ArCRZ1.** On encounter with the pathogen, plant generates various stresses around the site of pathogen recognition; a common stress is oxidative stress. The perception of these stresses by the pathogen, results in the upregulation of cytosolic $Ca^{2+}$ within the pathogen. Here, in *A. rabiei*, increased calcium level, sensed by calmodulin mediates the activation of calcineurin. Activated calcineurin dephosphorylates cytoplasmic inactive ArCRZ1 (phosphorylated form of ArCRZ1 remains inactive). Dephosphorylation of ArCRZ1 results in its translocation to the nucleus where it regulates the expression of *ArF-BAR* and other genes to cope with stress. Further, during polarized growth this ArF-BAR gets localized to hyphal tip where it leads to generation and stabilization of membrane curvature which is crucial for endocytosis and helps in the secretion of effector proteins required for fungal virulence. Additionally, ArF-BAR protein mediates actin organization and helps in septa formation.

during the early stages of pathogenic development [7]. However, the slow growth of the mutant (Figs 3 and S8) may be the possible cause of the reduced virulence by itself. We observed that the symptoms in plant bioassay (Figs 2 and S6) are severe in case of WT. The Δ*arf-bar* mutant showed only small or almost no lesions on plants. These observations clearly indicated that such severity is not merely due to slow growth of hyphae. Therefore, we hypothesized that the knockout mutant may have impaired in ability to secrete effectors of *A. rabiei*. Fungal pathogens secrete an array of effector proteins to counter host immunity and promote infection process [50]. Δ*arf-bar* mutant strain was found defective in secretion of a secretary signal sequence containing effector, Ar93 (Fig 5D). This experiment clearly shows that cumulative effect of both the slow growth and effector secretion are the primary cause of severe virulence defects in Δ*arf-bar* mutant. Additionally, some of the fungal effectors can change the structural and functional conformations as well as the localization of host target proteins that are key factors in host ROS production [51]. Pathogen survival under host-generated ROS is important for fungal growth and virulence. We also observed high transcript accumulation of *ArF-BAR* under oxidative stress conditions (Fig 1B). However, it will be interesting to decipher the entire mechanism of pathogen survival correlating all the components together.

As discussed above, during host-pathogen interactions, the key to successful pathogenesis is to overcome the rigid defense responses of the host. Collectively, the first challenge the

pathogen encounters is an oxidative burst at the site of infection, which initiates various signaling cascades in the pathogen that aid its survival. Calcium, an essential secondary messenger, mediates one such signaling cascade [52]. In response to stimuli, the cytosolic $Ca^{2+}$ concentration increases [53] and modulates various $Ca^{2+}$-binding proteins such as calmodulin. The $Ca^{2+}$ and calmodulin complex activates calcineurin. In many eukaryotes, calcineurin is known to regulate the activity of CRZ1, which is usually localized in the cytosol in phosphorylated form. Upon activation, CRZ1 gets relocated to the nucleus. In pathogenic fungi, many of the CRZ1-dependent targets, such as those involved in the maintenance of cell wall integrity, thermo-tolerance, cation homeostasis, azole tolerance, and hyphal growth have been identified [54]. Here, our study assumes functional significance for plant infection as we identified an F-BAR protein that is regulated by transcriptional network. We have identified the calcium-regulated CRZ1 transcription factor that regulates the transcription of F-BAR. These results, improves our understanding of the novel regulatory mechanism of endocytosis in filamentous fungi, where *ArF-BAR* functions downstream of ArCRZ1.

The development of septa is an important event during hyphal differentiation that is required for the formation of sexual structures and asexual spores [53]. Septation is comparable to cytokinesis that additionally includes cell separation. A cascade of events is involved in septum biogenesis, which includes assembly of the contractile actomyosin ring (CAR), plasma membrane ingression and cell wall constriction [55]. In fission yeast, Cdc15 and Imp2, and in budding yeast, Hof1 are the major F-BAR domain-containing proteins implicated in the formation of the CAR, and the primary and secondary septa during cytokinesis [56,57]. Additionally, *U. maydis* Cdc15 is also implicated in septin ring formation [16]. Therefore, the localization of ArF-BAR along the septum ring indicates that endocytosis is one of the pathways responsible for regulating the development of the septa. Septation is initiated at the definitive size of the hyphae [58]. With the initial recognition and establishment of disease in the host, the pathogen needs to proliferate at an enormously increased rate. Therefore, maintaining proper hyphal architecture and polarity is fundamental for pathogenesis that demands rapid coordinated internalization and recycling events [59]. The fungal mutant Δ*arf-bar*, deficient in ArF-BAR protein, revealed delay in septa formation and displayed decreased virulence in the absence of proper hyphal structure. In summary, we propose that the ArF-BAR protein of *A. rabiei* has the potential to interactively affect the hyphal growth and virulence of filamentous fungus. This evolving model provides mechanistic insight into the role of a membrane scaffolding protein in the process of endosome trafficking in fungal pathogenesis. In turn, this provides many additional potential targets for the development of effective and durable strategies to control AB fungal disease. Therefore, the observations of the present study in context to the intracellular trafficking, during the early stages of plant-pathogen interactions, have broad relevance for shaping disease-control strategies. These findings may also be helpful for the disease control of animal-infecting fungal pathogens such as *A. fumigatus* and *C. albicans*. Thus, further studies will be directed to characterize the interacting partners influenced by ArF-BAR during endosome formation. Further, the understanding of its regulatory mechanism would help in scrutinizing the molecular and cellular basis of disease development, which would subsequently help develop disease-control strategies for filamentous fungi.

## Materials and methods

### Fungal strains and growth conditions

A virulent isolate of *Ascochyta rabiei* (ArD2; ITCC No. 4638) was procured from IARI, New Delhi. The single spore culture of this mating type 2 isolate was generated and maintained as wild type fungi for research work. The WT and its derivative fungal strains (S2 Table) were

maintained on potato dextrose agar (PDA; Difco Laboratories, pH 5.2–5.5) at 22˚C for 10–15 days to assess the growth pattern and colony characteristics [26]. The fungus was routinely subcultured on PDA plates supplemented with chickpea extract to maintain virulence. To determine the vegetative growth pattern of fungal mycelia in response to oxidative stress, PDA plates supplemented with menadione (250 μM and 500 μM; Sigma-Aldrich, USA) or $H_2O_2$ (2 mM; Sigma-Aldrich, USA). The conidial suspensions of 10 μL ($1x10^3$ conidia/mL) were inoculated at the centre of PDA plates for growth assay. After 10 days of incubation under optimum growth conditions, the diameters of fungal colonies were measured using ImageJ software. Three independent biological experiments were performed with three technical replicates each time.

## RNA extraction and expression analysis

Total RNA was extracted from 5–6 days old fungal mycelia grown in potato dextrose broth (PDB; Difco Laboratories, USA) or from plant tissues inoculated with *A. rabiei* wild-type (WT) using TRIzol reagent (Invitrogen, USA). The isolated total RNA was subjected to DNase1 (Promega, USA) treatment and subsequently used for first-strand cDNA synthesis using SuperScript IV reverse transcriptase (ThermoFisher Scientific). Targeted gene expression was determined by qRT-PCR with ABI7900 (Applied Biosystems, USA), using SYBR Green PCR master mix (Applied Biosystems, USA). Relative expression of fungal genes was calculated after normalization with elongation factor α (*ArEFα*; ST47_g4052) using this $2^{-\Delta\Delta ct}$ method [60]. The data were analysed from three biological replicates each having three technical replicates.

## Site-directed mutagenesis

The mutations at required sites of *ArF-BAR* were achieved by PCR amplification of pET28a (+):*ArF-BAR* clone with pre-designed primers containing mutations of interest. Mutagenesis was performed using the QuikChange II Site-directed mutagenesis kit (Agilent, USA). Web-based Quik Change Primer Design tool (www.agilent.com/genomics/qcpd) was used to design primers. The presence of mutations in clones was confirmed by Sanger sequencing.

## Targeted gene knockout and complementation in *A. rabiei*

Homologous gene replacement with *hph* cassette strategy was used to generate knockout (KO) constructs for *A. rabiei* genes. Genomic DNA was isolated from 5-day grown PDB culture of *A. rabiei* using GenElute Plant Genomic DNA miniprep kit (Sigma-Aldrich, USA). The 5′ flanking genomic sequences of *ArF-BAR* were amplified from *A. rabiei* genomic DNA using primer pairs of ArF-BARKOif5F and ArF-BARKOif5R while the 3′ flanking sequences were amplified using Ar72KO3 and Ar72KO4 (S3 Table). These amplified 5′ and 3′ flanking sequences were cloned sequentially into pGKO2 vector at *Kpn*I/*Pst*I and *Bam*HI/*Eco*RI sites, respectively. The cloned *ArF-BAR* gene replacement cassette of ~3.4 kb, including 5′ and 3′ flanking sequence along with *hph*, was amplified using primer pair ArF-BARKOif5F and Ar72KO4, and transformed into *A. rabiei* protoplasts. The *A. rabiei* protoplast transfection was performed as described earlier [61], with some minor modifications. The putative transformants were selected on a PDA plate supplemented with 50 μg/μL hygromycin. To generate the complementation constructs of *ArF-BAR* and its mutated (mut1 and mut2) versions, about 4.4 kb DNA fragment having the *ArF-BAR* native promoter, ORF region, and TrpC terminator was amplified and cloned into pBIF2 vector (Bacterial selection- kanamycin; Fungal selection-G418) at *Eco*RI and *Hind*III sites. These three constructs of *ArF-BAR* were independently transformed into Δ*arf-bar* mutant strain by *Agrobacterium tumefaciens*-mediated

transformation (ATMT) [27]. Similar to generation of *ArF-BAR* knockout and complementation mutant, *A. rabiei ArCRZ1* gene knockout mutant and complementation strains were also developed. *A. rabiei ArCRZ1* gene knockout mutant and complementation strains were developed with oligonucletides enlisted in S3 Table.

### Gene knockout confirmation by PCR and Southern blot

The single spore culture of putative KOs selected on hygromycin was initially screened by genomic PCR. To check the proper recombination of deletion construct at desired genomic region, a primer set binding at position 5′ to the homologous recombination region and TrpC promoter was used for gene 5′ upstream region while a primer set binding at *Hph* gene and position 3′ to the homologous recombination region was used for gene 3′ downstream region (S3 Table). The putative transformants for complementation strains were also screened by genomic PCR. The PCR positive Δ*arf-bar*, Δ*arcrz1* mutants (KOs), and complemented KO strains were further verified by Southern blot. The genomic DNA of WT, Δ*arf-bar* and Δ*arcrz1* mutants was digested with *Eco*RI enzyme while genomic DNA was digested with *Eco*RI and *Hind*III for the complementation strains (Δ*arf-bar/ArF-BAR*, Δ*arf-bar/ArF-BAR*^mut1^, Δ*arf-bar/ArF-BAR*^mut2^, and Δ*arcrz1/ArCRZ1*). The digested DNA was separated along with λ DNA/*Hind*III marker (ThermoFisher Scientific) and blotted to a membrane followed by hybridization with radioactive probe prepared using the random primers labeling NEBlot kit (New England Biolabs, USA). The band detection was carried out using Typhoon phosphor imager (GE Healthcare, USA).

### Plant infection assay

Two-weeks-old susceptible chickpea (Pusa 362) plants grown in plant growth chambers under controlled conditions (D/N temperature: 24˚C/18˚C; Relative Humidity: 80%; light intensity 250 μE/m$^2$/s; D/N light duration: 14/10 h) were used for infection assays. Conidial suspensions of *A. rabiei* strains were collected separately from a 20-days-old PDA plate grown culture. Two-weeks-old plants were spray inoculated with conidial suspensions diluted to 2x10$^6$conidia/mL. Plants were again kept under optimum conditions. Disease lesions were examined 5–7 days after spray inoculation.

### *In vitro* protein purification

The ORFs of desired genes were amplified from cDNAs of the interest, cloned in pET28a(+) and transformed into *E. coli* BL21-CodonPlus (DE3)-RIPL cells. Protein expression was induced with 0.5 mM Isopropyl β-D-thiogalactoside (IPTG) for 6 h at 23˚C. Bacterial pellet was lysed in buffer [500 mM NaCl, 50 mM NaPO$_4$ (pH 8.0), 10 mM Imidazole, 1mM β-Mercaptoethanol, 1 mg/mL lysozyme and 1mM Phenylmethylsulfonyl fluoride (PMSF)] by incubating 30 min on ice, followed by sonication (10 cycle, 70% amplitute). The cell lysates were precipitated followed by 0.45 μm filtration. The cleared lysate was incubated with Ni-NTA resin for 30 min at 4˚C. The His-tagged fusion proteins were purified using a Ni-NTA column (Bio-Rad, USA). Proteins were eluted in elution buffer [500 mM NaCl, 50 mM NaPO$_4$ (pH 8.0), 10% glycerol and 200 mM imidazole]. The quality and quantity of eluted protein were checked by SDS/PAGE and Bradford assay, respectively. The proteins were dialyzed in compatible buffers, as per experiment, followed by concentration.

## Liposome preparation and tubulation assay

Liposomes were prepared using 70% Phosphatidylethanolamine (POPE), 20% Phosphatidyl-choline (POPC), and 10% Rhodamine B-conjugated PE (Echelon Biosciences, USA). In an amber glass vial, all the lipids were initially dissolved in chloroform: methanol (65:35; v/v) mixture and vials were kept under liquid nitrogen for 10 min before being immediately subjected to vacuum desiccation/lyophilisation for 2 h at 60 mTorr. The lipids were hydrated with buffer [25 mM Tris-HCl (pH 6.8) and 100 mM NaCl] and subjected to three freeze-thaw cycles of 5 min each at 68°C and liquid nitrogen. Extrusion was performed at 68°C on a pre-heated mini extruder (Avanti Polar Lipids, USA). The prepared liposomes were diluted as desired and immediately proceeded for tubulation assay. Before use, the purified proteins were subjected to 100,000 g centrifugation for 20 min at 4°C to remove the aggregates. To examine tubule formation, the mixed liposome and protein samples were analysed in Lumox 24-well plate (Millipore, USA) by live-cell imaging on Axio Examiner.Z1 (Zeiss microscope).

## Liposome co-sedimentation assay

The purified fusion proteins were pre-centrifuged, before the assay, at 100,000 g for 15 min to remove protein aggregates. Protein from the supernatant was mixed with freshly prepared synthetic liposomes with gentle tapping. Ultracentrifugation was performed at 100,000 g for 15 min at 4°C and the supernatant and the pellet were carefully separated. The supernatant was mixed with 1:1 loading buffer while the pellet was re-suspended in 2x loading buffer. Samples were analysed on Coomassie stained SDS-PAGE.

## Yeast two-hybrid assays

The interactions of various combinations between ArF-BAR, ArActin, ArWASP, and ArF--BAR domains in yeast cytoplasm were examined using the split-ubiquitin based DUALhunter system (Dualsystems Biotech). The ORFs were cloned at *Not*I and *Asc*I restriction sites or by LR clonase II into pGDHB1 and pGPR3-N vectors. The cloned plasmids were co-transformed along with necessary controls into NMY51 strain using the EZ-Yeast transformation kit (MP Biomedicals, USA) and plated on SD/-L/-W plates. The plating of yeast clones on required synthetic media to check protein-protein interactions in yeast was done as described previously [22]. All the interactions were verified by three independent experiments.

## Effector secretion assay

For total fungal protein extraction, fungal mycelia were grown in PDB at 22°C, 120 rpm for 7 days and harvested by filtering through three layers of sterile Mira cloth (EMD Millipore Corp, Germany). Tissues were snap freezed in liquid nitrogen and stored in -80°C for future use. The fungal tissue was grounded to fine powder and resuspended in Tris-glycine buffer pH 8.3 (3g Trizma, Sigma-Aldrich, USA and 14.4 g Glycine, Sigma-Aldrich, USA dissolved in 1 L MQ). Resuspension was then centrifuged at 16 x 1000g for 40 min at 4°C. The supernatant and cell lysate was collected separately. The total protein from the supernatant was separated on SDS-PAGE followed by immune blot with anti-Flag Ab (Cohesion biosciences, Singapore). For the extraction of secreted proteins, 7 days old grown mycelia in PDB was treated with 250 μM menadione (Sigma-Aldrich, USA), to mimic host induced oxidative stress condition. Axenic culture filtrate (CF) was collected and filtered through three layered Mira cloth (EMD Millipore Corp, Germany) to separate fungal mycelia. Further, CF was sequentially filtered with 0.45 and 0.22 μm Durapore PVDF membrane filters (Sigma-Aldrich, USA). Filtered CF was then concentrated using 3 kDa Amicon Ultra-15 Centrifugal Filter Units (Merck, USA).

The concentrated secreted proteins were then separated on SDS-PAGE followed by immuno-blot with anti-Flag Ab.

## Actin polymerization assay

The actin polymerization modulation activity of the proteins was checked using Actin Polymerization Biochem Kit (Cytoskeleton, USA), using the manufacturer's instruction. Freshly solubilized components- 13 nM Arp2/3 protein complex and 15 nM WASp-VCA domain-GST purified (Cytoskeleton, USA) along with freshly purified ArF-BAR protein in concentrations of 200, 400, and 600 nM were used. The reaction was carried out in an opaque 96-well plate and kept in the dark. The actin polymerization rate was recorded by monitoring the pyrene fluorescence signals using CLARIO star plate reader (BMG Labtech, Germany) with the following settings; slow kinetics, 60 sec intervals, $\lambda_{ex} = 360\pm15$ nm and $\lambda_{em} = 420\pm20$ nm.

## Electrophoretic mobility shift assay (EMSA)

The His-tagged ArCRZ1 (*His-ArCRZ1*) expression construct was generated and recombinant protein was purified. The wild type and mutated CDRE DNA fragments were assembled by annealing oligonucleotide pairs in a thermal cycler by heating at 95˚ C for 5 min followed by cooling at RT for 15 min. End labelling of DNA fragments was performed by [$\gamma^{32}$P] dATP (Board of Radiation & Isotope Technologies, BRIT, India) and Polynucleotide Kinase (Thermo Fisher Scientific, USA). Additionally, for competition assays, two complementary oligonucleotides were annealed at equimolar concentration. Purified His-ArCRZ1 protein was incubated with 10 ng of labelled DNA fragment in the presence of 1 mg of poly-deoxy-inosinic-deoxy-cytidylic acid [poly (dI-dC)] and 1X binding buffer (15 mM HEPES (pH 7.6), 0.2 mM MgCl$_2$, 35 mM KCl, 1 mM DTT and 1% glycerol) in a reaction volume of 30 μL for 25 min at room temperature. DNA loading dye was used to terminate the reaction. The competitive assays were performed using 50, 100, and 200 times of specific fragments in excess. To identify the relative binding, the complexes were resolved on 6% native PAGE, dried, and autoradio-graphed on X-ray films.

## Microscopic methods

**Sample preparation for microscopic analysis.** Conidia from the fungal strains were harvested in 1 mL sterile distilled water from 15-day-old fungal mycelia grown on PDA plates. The conidial suspension was filtered through Mira cloth. Ten microlitres ($1\times10^6$ conidia/mL) of suspension was kept on a sterile glass coverslip and allowed to grow under the optimal conditions for 24 h in dark. To investigate the localization of chimeric proteins in fungal hyphae, growing on chickpea stem peel, the conidia were allowed to grow for 36 h under optimum conditions of infection. The grown hyphae samples were then used for confocal laser scanning microscopy.

**Confocal laser scanning microscopy.** For microscopic studies, TCS SP5 and TCS SP8 confocal laser scanning microscopes (Leica Microsystems, Germany) were used. For subcellular localization, conidia were harvested from transgenic fungal strains expressing fluorescent-tagged proteins. The Z-stacked images with 1 μm step size were acquired using a high-resolution CCD camera. For calcofluor-white (CW), the hyphae grown on the glass slide were incubated for 10 min in the CW solution (Sigma-Aldrich, USA). After incubation, the stained hyphae were rinsed with PBS (pH 7.4) followed by rinsing with sterile distilled water, before image acquisition. For FM4-64 uptake, 1 mL of harvested conidial suspension was centrifuged at 2,500 g, washed twice with sterile distilled water and then allowed to germinate on glass slides. The aqueous solution (10 μM) of FM4-64 dye (Invitrogen, USA) was added directly to

the fungal mycelia. After 10 min incubation, FM4-64 dye was rinsed from slides thoroughly before imaging fungal hyphae. Images were captured with a TCS SP8.

For host penetration assay, infected chickpea leaves (WT and Δ*arf-bar*) were placed in 100% ethanol for 48 h to undergo bleaching for the complete removal of chlorophyll. Subsequently, leaves were incubated for 4 h in 10% KOH at RT followed by washing 4–5 times in phosphate buffer saline (PBS, pH 7.4). The processed leaves were then stained with chitin specific dye WGA-AF 488 (Invitrogen, USA). The leaf samples were rinsed in PBS (pH 7.5) before microscopic visualization. Confocal images were captured with a TCS SP5 confocal microscope.

For sub-cellular localization of ArCRZ1::EYFP under oxidative stress conditions, fungal conidia expressing chimeric protein were isolated and allowed to grow for 12 h. The hyphae were exposed to $CaCl_2$ (200 mM, Sigma-Aldrich, USA) for 1 min prior to microscopy. To assess the involvement of calcineurin in nuclear localization of ArCRZ1::EYFP, the hyphae were exposed to 5 μg/μL FK506 (Sigma-Aldrich, USA), 5 min prior to the addition of $CaCl_2$. The confocal images were acquired with a TCS SP5.

For subcellular localization of ArCRZ1::EYFP during infection, susceptible plants were challenged with conidia expressing ArCRZ1::EYFP. The images were acquired after 48 h using TCS SP5.

**Quantification and statistical analysis.** Quantification analysis of relative fluorescent intensity, lesion size, radial diameter, and the distance of septa from the hyphal tip were analysed by ImageJ/Fiji software. To calculate the significance of means/differences between two groups, Student's t-test and one-way ANOVA followed by Tukey test between multiple groups were performed using GraphPad Prism 6. Significance was accepted at $p < 0.05$, as noted in the text of legends. Replicates are indicated in the legends.

**Bioinformatic analysis.** All the gene and protein sequences were acquired from NCBI server (http://www.ncbi.nlm.nih.gov). Multiple sequence alignment was performed by Praline search (http://www.ibi.vu.nl/programs/pralinewww/). Phylogenetic analysis was performed with MEGA7.0.21 software. Distinctive domain organization of the protein was determined by SMART search (http://smart.embl-heidelberg.de/). The putative TFs binding sites were identified by YEASTRACT-DISCOVERER Database (http://yeastract.com). The theoretical pI and molecular weight of the chimeric proteins were determined by Expasy compute pI/Mw tool (http://web.expasy.org/cgi-bin/compute_pi/pi_tool). Protein IDs of protein sequences used in this study are: ArF-BAR: KZM20872.1, ArRab5: KZM19760.1, ArRab7: KZM26450.1, ArActin: KZM21342.1, ArWASP: KZM19192.1, ArCRZ1: KZM25117.1, and ArSNC1: KZM18806.1.

## Supporting information

**S1 Fig. Maximum likelihood phylogeny of F-BAR sequences.** Sequences used: *Ascochyta rabiei*, *Parastagonospora nodorum*, *Alternaria alternate*, *Bipolaris zeicola*, *Bipolaris victoriae*, *Bipolaris oryzae*, *Bipolaris sorokiniana*, *Bipolaris maydis*, *Sclerotinia sclerotiorum*, *Botrytis cinerea*, *Marssonina brunnea*, *Histoplasma capsulatum*, *Blastomyces dermatitidis*, *Blastomyces gilchristii*, *Emmonsia crescens*, *Paracoccidioides brasiliensis*, *Uncinocarpus reesii*, *Coccidioides immitis*, *Aspergillus nidulans*, *Aspergillus rambellii*, *Aspergillus ochraceoroseus*, *Aspergillus kawachii*, *Aspergillus lacticoffeatus*, *Aspergillus oryzae*, *Aspergillus flavus*, *Aspergillus parasiticus*, *Aspergillus nomius*, *Aspergillus terreus*, *Aspergillus fumigatus*, *Aspergillus lentulus*, *Aspergillu sclavatus*, *Aspergillus fischeri*, *Trichoderma reesei*, *Cordyceps confragosa*, *Fusarium oxysporum*, *Fusarium verticillioides*, *Fusarium fujikuroi*, *Fusarium graminearum*, *Fusarium pseudograminearum*, *Fusarium langsethiae*, *Neonectria ditissima*, *Metarhizium anisopliae*, *Metarhizium acridum*, *Pyricularia grisea*, *Pyricularia oryzae*, *Verticillium dahlia*, *Neurospora*

*crassa*, *Drosophila melanogaster*, *Schizosaccharomyces pombe*, *Phytophthora sojae*, *Phytophthora graminis*, *Phytophthora striiformis*, *Ustilago maydis*, *Rhizopus delemar*, *Cryptococcus neoformans*, *Phytophthora infestans*, *Yarrowia lipolytica*, *Kluyveromyces lactis*, *Naumovozyma castellii*, *Saccharomyces cerevisiae*, *Candida glabrata*, *Eremothecium gossypii*, *Clavispora lusitaniae*, *Meyerozyma guilliermondii*, *Candida parapsilosis*, *Candida tropicalis*, *Candida dubliniensis*, *Candida albicans*, *Ganoderma lucidum*, and *Homo sapiens*. The multiple sequence alignment of protein was performed by PROMALS3D software and the phylogeny was constructed using a software MEGA7.0.21. The bootstrap values, derived from 1000 iterations, validated the obtained phylogeny. (TIF)

**S2 Fig. Conserved nature of ArF-BAR protein.** Multiple sequence alignment showing the conservation of ArF-BAR protein with BZZ1p of *S. cerevisiae*, Cdc42-interacting protein 4 (CIP4) of *Drosophila* and Syndapin proteins of *Drosophila*. Colour code for sequence conservation varies from blue (least conserved) to red (highly conserved). The alignment of the protein is determined by Praline software using default parameters. The black box marks the presence of positively charged residues of F-BAR domain. Asterisk (*) represents the residues in C1 domain of protien Kinase C1 (PKC1), required for interaction with DAG. (TIF)

**S3 Fig. Expression and purification of ArF-BAR protein in *E. coli*.** (A) His purification of bacterially expressed ArF-BAR protein. Analysis of the purification of recombinant ArF-BAR as shown by SDS-PAGE. (B) Intense tubular network in synthetic liposomes is formed after 30 min incubation with purified recombinant ArF-BAR protein. Inset showing the enlarged view of dense tubular network originating from a liposome. (C- D) His purification of ArF-BAR$^{mut1}$ and ArF-BAR$^{mut2}$. The protein was visualised by Coomassie Brilliant Blue staining. (UI- crude extract of un-induced samples after centrifugation; I- crude extract of induced samples after centrifugation; FT- flow-through fraction of the Nickel chelating resin column; W5- 5$^{th}$ wash fraction of the Nickel chelating resin column; E1, E2 and E3- eluate fractions of the Nickel chelating resin column showing the purified ArF-BAR protein). Protein standards are shown (M) and their masses are indicated in kDa. (TIF)

**S4 Fig. Southern hybridization confirmation of the successful replacement of *ArF-BAR* gene with hygromycin cassette and complementation.** (A) The schematic representation of *A. rabiei* knockout mutant generation by the homologous recombination approach to obtain targeted *ArF-BAR* gene deletion mutants (Δ*arf-bar*). The bar represents the genomic region used to generate probe for Southern confirmation. (B) The representative Southern blot confirming successful *ArF-BAR* gene deletion (Δ*arf-bar*), with a single integration of *hph* at replacement site. Along with the confirmation of *ArF-BAR* complementation in Δ*arf-bar*, followed successful generation of Δ*arf-bar/ArF-BAR*$^{mut1}$ and Δ*arf-bar/ArF-BAR*$^{mut2}$ complementation. (TIF)

**S5 Fig. The schematic representation of constructs used to generate Δ*arf-bar* mutant complemented strains.** (A, B, C) Constructs used to generate different Δ*arf-bar* mutant complemented strains (Δ*arf-bar/ArF-BAR*, Δ*arf-bar/ArF-BAR*$^{mut1}$, and Δ*arf-bar/ArF-BAR*$^{mut2}$) under the control of native promoter of *ArF-BAR* gene. (TIF)

**S6 Fig. Disease symptoms on AB susceptible chickpea plants after 10 days post inoculation.** The susceptible plants inoculated with conidia of *A. rabiei* (WT), Δ*arf-bar/ArF-BAR*, and

Δ*arf-bar/ArF-BAR*<sup>mut2</sup>, showed severe disease symptoms with the increasing duration after inoculation. Δ*arf-bar* and Δ*arf-bar/ArF-BAR*<sup>mut1</sup> challenged plants showed no or very mild symptoms after 10 dpi.
(TIF)

**S7 Fig. Evaluation of chickpea host cells penetration by *A. rabiei* (WT) and Δ*arf-bar* strains growing hyphae.** (A, B) Confocal images showing the depth of penetration 48 hpi by *A. rabiei* (WT) and Δ*arf-bar* strains, respectively, in AB susceptible chickpea leaves. Fungal hyphae were stained with WGA-488 for visualization, prior to microscopy. The *Z*-stacked images were acquired till 23 μm depth, starting from the surface of the leaves. The image is the representation of maximum projections of all the Z-stacks. Scale bar = 5 μm. (C) The bar graph, representing mean and SD, shows the difference in ability to penetrate within the host by *A. rabiei* (WT) and Δ*arf-bar*. The results were analysed using Student's t-test one tailed compared to its control (*$p$ = 0.0079).
(TIF)

**S8 Fig. Growth inhibition in colony morphology of Δ*arf-bar* mutant and ArF-BAR complementation strains in the presence of various oxidative stresses.** The *A. rabiei* (WT), Δ*arf-bar* mutant and mutant complemented strains, observed 10 days after incubation at 22˚C. For oxidative stress, the PDA was supplemented with 250 μM and 500 μM menadione and 2 mM $H_2O_2$. Strains of Δ*arf-bar* and Δ*arf-bar/ArF-BAR*<sup>mut1</sup> exhibited more growth inhibition towards oxidative stress condition as compared to WT.
(TIF)

**S9 Fig. Distribution of ArF-BAR::EYFP during *in planta* infection.** Confocal micrographs showing the punctate distribution of ArF-BAR::EYFP during host infection. The representative image is the maximum intensity projection of all the Z-stack images with 0.5 μm step size, acquired after 48 hpi of susceptible chickpea with fungal conidia expressing chimeric ArF-BAR::EYFP. Scale bar = 5 μm (n = 12).
(TIF)

**S10 Fig. Germination of fungal conidia of *A. rabiei* and Δ*arf-bar* on hydrophobic surface.** Difference in growth pattern on hydrophobic surfaces was slightly observed and microscopic photographs (20X) were taken after 12 h of conidial spread. Scale bar = 50μm.
(TIF)

**S11 Fig. Homodimerization of F-BAR indicating conserved nature.** Split-Ubiquitin based Y2H system was used to determine the homodimerization between ArF-BAR proteins. Plates were photographed after 48 h of yeast growth. Strong positive interaction between two ArF-BAR proteins was reflected with the growth on QDO (SD/-L/-W/-A/-H) media and X-gal overlay assay to check the activation *LacZ* gene.
(TIF)

**S12 Fig. F-BAR domain of ArF-BAR is not the direct member to interact.** (A) The yeast two-hybrid result showing the positive interaction of ArF-BAR protein with ArWASp. (B) F-BAR domain (amino acids 1–325) of ArF-BAR protein [F-BAR(<sub>ArF-BAR</sub>)] failed to interact with ArWASp in Y2H system. Plates were photographed 48 h after yeast spotting. The interaction was confirmed by three independent replicates.
(TIF)

**S13 Fig. Characterization of ArCRZ1 and confirmation of gene knockout and complementation.** (A) Schematic representation of domain organisation of ArCRZ1 protein. (B) The

schematic map showing the homologous recombination based knockout approach used for targeted *ArCRZ1* gene deletion mutant (Δ*crz1*) strain generation. (C) Schematic representation of Δ*arcrz1/ArCRZ1* complementation construct under the native promoter of *ArCRZ1*. The genomic region used to generate probe for confirmation by Southern hybridization is being highlighted. (D) The Southern blot result confirmed successful *ArCRZ1* gene deletion (Δ*arcrz1*), with single integration of *hph* at replacement site along with the complementation confirmation of *ArCRZ1* in Δ*arcrz1* mutant.
(TIF)

**S1 Table. YEASTRACT result for the putative transcription factors bindings on the upstream regulatory sequences of *ArF-BAR*.**
(DOCX)

**S2 Table. *Ascochyta rabiei* strains used in this study.**
(DOCX)

**S3 Table. List of oligonucleotides used in this study.**
(DOCX)

**S1 Data. Supplementary Data.**
(XLSX)

## Author Contributions

**Conceptualization:** Vimlesh Kumar, Praveen Kumar Verma.

**Data curation:** Ankita Shree, Kunal Singh, Kamal Kumar, Vimlesh Kumar, Praveen Kumar Verma.

**Formal analysis:** Praveen Kumar Verma.

**Funding acquisition:** Praveen Kumar Verma.

**Investigation:** Manisha Sinha, Ankita Shree, Kunal Singh, Shreenivas Kumar Singh.

**Methodology:** Manisha Sinha, Ankita Shree, Kunal Singh, Shreenivas Kumar Singh, Praveen Kumar Verma.

**Project administration:** Praveen Kumar Verma.

**Resources:** Kamal Kumar, Shreenivas Kumar Singh.

**Supervision:** Kunal Singh, Kamal Kumar, Vimlesh Kumar, Praveen Kumar Verma.

**Validation:** Manisha Sinha, Ankita Shree.

**Visualization:** Manisha Sinha, Ankita Shree, Praveen Kumar Verma.

**Writing – original draft:** Manisha Sinha, Ankita Shree, Vimlesh Kumar, Praveen Kumar Verma.

**Writing – review & editing:** Ankita Shree, Kamal Kumar, Praveen Kumar Verma.

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
