## [Decision Letter · Decision Letter 0]

8 Nov 2020

Dear Dr Verma,

Thank you very much for submitting your Research Article entitled 'Modulation of fungal virulence through CRZ1 regulated F-BAR-dependent actin remodeling and endocytosis in chickpea infecting phytopathogen Ascochyta rabiei' to PLOS Genetics. Your manuscript was fully evaluated at the editorial level and by independent peer reviewers. The reviewers appreciated the attention to an important problem, but raised some substantial concerns about the current manuscript. Based on the reviews, we will not be able to accept this version of the manuscript, but we would be willing to review again a much-revised version. We cannot, of course, promise publication at that time.

If you decide to revise the manuscript for further consideration at PLOS Genetics, please aim to resubmit within the next 60 days, unless it will take extra time to address the concerns of the reviewers, in which case we would appreciate an expected resubmission date by email to plosgenetics@plos.org.

[LINK]

We are sorry that we cannot be more positive about your manuscript at this stage. Please do not hesitate to contact us if you have any concerns or questions.

Yours sincerely,

Eva H. Stukenbrock, PhD

Associate Editor

PLOS Genetics

Gregory P. Copenhaver

Editor-in-Chief

PLOS Genetics

Three reviewers have provided detailed comments to your manuscript. Please note comments provided as attachments by reviewer 2 and 3. In a revised manuscript, all comments and suggestions by the reviewers should be taken into account.

Reviewer's Responses to Questions

**Comments to the Authors:**

Reviewer #1: Modulation of fungal virulence through CRZ1 regulated F-BAR-dependent actin remodeling and endocytosis in chickpea infecting phytopathogen Ascochyta rabiei --Manuscript Draft—

Review

The authors present data studying a host-pathogen system involving the necrorophic funbgus Ascochyta rabiei and the host plant chickpea.

The main findings are: Bin1/Amphiphysin/Rvs167 (BAR) domain-containing protein ArF-Bar is essential for virulence. ArF-BAR regulates endocytosis at the hyphal tip, localizes to the early endosomes, and is involved in actin dynamics. Mutants were generated and complemented.

Plant infection assays were conducted; the mutant was shown to to delay septation.

A further aspect was evidence that Ar Fis involved in tubule formation of membranes.

Genetic studies indicated that ArF expression is regulated by Crz1.

Comments:

The paper setup is ok and the microscopy, and figures are pretty decent.

The species name should include a mention of Didymella rabiei. Also the pathovar should be indicated.

L 23-25 The authors state „The loss-of- function of ArF-BAR results in delayed formation of first septa from the hyphal tip, crucial for host penetration and proliferation.“

Yet, a direct microscopic observation of appressorium formation/penetration oft he plant cuticula is not presented. Instead the paper turns to genetic regulation of Ar Fand identifies Crz1.

The importance of this regulation is not clear, however, as a role for Crz1 in virulence has not been determined.

Fig 1A: beyond a comparison in suppl figures S1 and S2 it is of more interest to see if the domain structure is also conserved. The ScBzz1/Las7 actually interacts with type I myosins…

Fig 1B: the authors show that expression of Ar Fis induced upon germination. Yet they state that this equals „transcript induction during pathogenesis“. I tend to disagree.

The mutant shows slow growth (Fig. 2). This could reduce virulence by itself as an indirect effect. This should be discussed more clearly.

Figure 2C shows that there are a few lesions formed in the plant virulence assay. Thus the authors should not state that ArF is essential for virulence, but required for full virulence as stated in L179+.

L172: „Typical AB disease symptoms were observed“ what is menat with AB?

L191+ Again, the authros state that F-BAR is indispensable for fungal pathogenicity. This is double ungood. First, it is dispensible, but required for full virulence; second, the authros make an extrapolation on fungal virulence in general that is not warranted. In light oft he results also the PKC domain is required for full virulence, a point the authors just dropped.

The genetic analysis is ok – yet, there may be issues with protein stability oft he mutants that at least need to be discussed.

L193-205: the observed effedt of reduced penetration could also be a consequence of the slower growth, this needs tob e discussed.

L206++: ArF contains a PKC domain. To me this hints to a cell wall remodeling role. Oxidative stress is not the first stress that comes to mind then but rather osmotic stresses.

L206-221: if the authors want to stress this the should move the data from supplemental to results. I suggest tob e more critical with these results. Stats need to include the general growth defect oft he mutant and recalculate the relative growth inhibition on top oft hat.

L225: how was transient expressino of YFP achieved. This is not clear from the methods section.

L238-245: a statistic on the distribution of septa in the hyphae seems warranted to see if the spacing is random, irregular or if the compartments are simply larger than in WT.

The YFP data would be improved by a co-staining of the actin cytoskeleton. This would show that ArF colocalizes with actin patches.

The colocalization data of ArF with Rab5 and not with Rab7 is interesting. The suppl material on Rab7 needs to be moved to the results next to Rab5.

Overall:

The paper comes with a ton of results. This is very nice. Yet, the authors also produce a ton of figures and suppl material that overwhelms and distracts from the novel findings.

The authors should better indicate what is known from ArF homologs and build on that. The knowns may go to supplemental and the novel stuff should be highlighted in the figures.

Reviewer #2: Manuscript: PGENETICS-D-20-01505

Title: Modulation of fungal virulence through CRZ1 regulated F-BAR dependent actin remodeling and endocytosis in chickpea infecting phytopathogen Ascochyta rabiei

Authors: Manisha Sinha, Ankita Shree, Kunal Singh, Kamal Kumar, Vimlesh Kumar, Praveen Kumar Verma

Recommendation: Major Review

In this work, Sinha and colleagues functionally characterize an F-BAR domain-containing protein of the chickpea pathogen Ascochyta rabiei. The corresponding gene was identified in a transcriptomic analysis of the response of this pathogen to oxidative stress. After confirming this feature through qPCR, the authors assessed the subcellular localization of ArF-BAR, determined its role in virulence, characterized the phenotype of the null mutant under different stress conditions and proposed that it has a role in endosome biogenesis and endocytosis, as well as in the correct formation of the actin cytoskeleton. Furthermore, they identified the transcription factor Crz1 as regulator of arf-bar expression and, consequently, carried out a characterization of the corresponding null mutant.

In my opinion, the work done is of interest for the readers of Plos Genetics. However, some points need to be addressed before I recommend its publication. A first issue is that, although the manuscript is well written, it needs improvement and rewriting of several sentences. Second is that, for a better characterization of ArF-BAR, the double point mutant should be generated and its phenotype compared with those of Mut1 and Mut2. Third, the apical localization of ArF-BAR should be tracked in more detail. Additional markers of exocytosis and the Spitzenkörper should be used with that aim. Four, how do the authors explain the difference in ArF-BAR localization in Figure 3A compared to that shown in Figure 4B? Is it only a matter of the Z-position? In addition, the authors should indicate the hyphal region plotted in the graph of panel 4B. Five, I recommend to tone down any mention to a hypothetic direct role of ArF-BAR in endocytosis. I would say that what the authors have shown is a defect in endocytosis in the null arf-bar mutant, and that not necessarily means that ArF-BAR is directly involved in early endosome biogenesis and endocytosis. Six, I would show the phenotype of the null crz mutant under stress conditions as a panel of Figure 7 and not as supplementary material. And finally, I recommend a more detailed description of some of the procedures used. I attach a pdf file of the manuscript with all my comments and corrections. Hope it will help.

Reviewer #3: review as attachment

**Have all data underlying the figures and results presented in the manuscript been provided?**

Reviewer #1: Yes

Reviewer #2: Yes

Reviewer #3: Yes

PLOS authors have the option to publish the peer review history of their article (what does this mean?). If published, this will include your full peer review and any attached files.

Reviewer #1: No

Reviewer #2: No

Reviewer #3: No

---

## [Decision Letter · Decision Letter 1]

23 Mar 2021

Dear Dr Verma,

Thank you very much for submitting your Research Article entitled 'Modulation of fungal virulence through CRZ1 regulated F-BAR-dependent actin remodeling and endocytosis in chickpea infecting phytopathogen Ascochyta rabiei' to PLOS Genetics.

The manuscript was fully evaluated at the editorial level and by independent peer reviewers. We ask you to modify the manuscript according to the review recommendations. Minor revision is requested before the manuscript can be accepted. Your revisions should address the specific points made by each reviewer.

[LINK]

Yours sincerely,

Eva H. Stukenbrock, PhD

Associate Editor

PLOS Genetics

Gregory P. Copenhaver

Editor-in-Chief

PLOS Genetics

The three reviewers evaluated your revised manuscript. Reviewer 1 has no further comments to your manuscript. Reviewer 2 and 3 have listed a few minor issues (please see attached pdfs) that must be dealt with before the manuscript can be accepted for publication. These mainly relates to minor changes that will improve the presentation of the study.

Furthermore, I here list a few minor typos and text issues that should be corrected:

L. 28: “..secretion of a candidate effector” (please add “a” to sentence).

L. 65: “for motor-dependent retrograde signaling, endosomes are involved…” (replace “they” with “endosomes”)

L. 82: End of line typo (“t)

L. 88: “…virulence of the filamentous fungus…”

L. 119: What is meant with “unigenes”. Please clarify what this term defines.

L. 146: It is confusing that the ArF-BARmut2 strain is mentioned here as it is only explained further down what it is. You may either remove the mentioning of this particular mutant here, or you may add “(see below)” .

L. 170: “..with lipid membranes”

L. 180: “..To assess the effect…”

L. 260: Please remove “However, “. I believe this should be “The localization pattern…”

L. 325: “hypothesized the existence of…”

L. 326: type of first “.”

L. 433: “we propose that ArF-BAR contributes to endocytosis….”

L. 456: “..may be the possible cause…”

Reviewer's Responses to Questions

**Comments to the Authors:**

Reviewer #1: no

Reviewer #2: The authors have upgraded the manuscript and have addressed most of the comments of the reviewers. However, after reviewing this second version of the manuscript, I have still found several points and sentences that need rewriting. This should be addressed before acceptance. Besides that, I only have minor comments. First, at some points, I wonder if the number of replicates analyzed by the authors is enough. Second, I still recommend to move the figure showing the phenotype of the null crz1 strain to the main text. And third, I still recommend to give quantitative data, SD and p values in the main text. It is done in some cases but I believe that this should be the general rule.

Taking everything into consideration, my recommendation is Minor Review. I attach a pdf copy of the manuscript with all my comments and corrections.

Reviewer #3: Review uploaded

**Have all data underlying the figures and results presented in the manuscript been provided?**

Reviewer #1: None

Reviewer #2: Yes

Reviewer #3: None

PLOS authors have the option to publish the peer review history of their article (what does this mean?). If published, this will include your full peer review and any attached files.

Reviewer #1: No

Reviewer #2: No

Reviewer #3: No

---

## [Editor Report · Decision Letter 2]

13 Apr 2021

Dear Dr Verma,

We are pleased to inform you that your manuscript entitled "Modulation of fungal virulence through CRZ1 regulated F-BAR-dependent actin remodeling and endocytosis in chickpea infecting phytopathogen Ascochyta rabiei" has been editorially accepted for publication in PLOS Genetics. Congratulations!

Yours sincerely,

Eva H. Stukenbrock, PhD

Associate Editor

PLOS Genetics

Gregory P. Copenhaver

Editor-in-Chief

PLOS Genetics

Comments from the reviewers (if applicable):

**Data Deposition**

http://datadryad.org/submit?journalID=pgenetics&manu=PGENETICS-D-20-01505R2

**Press Queries**

---

## [Editor Report · Acceptance letter]

12 May 2021

PGENETICS-D-20-01505R2 

Modulation of fungal virulence through CRZ1 regulated F-BAR-dependent actin remodeling and endocytosis in chickpea infecting phytopathogen Ascochyta rabiei 

Dear Dr Verma, 

We are pleased to inform you that your manuscript entitled "Modulation of fungal virulence through CRZ1 regulated F-BAR-dependent actin remodeling and endocytosis in chickpea infecting phytopathogen Ascochyta rabiei" has been formally accepted for publication in PLOS Genetics! Your manuscript is now with our production department and you will be notified of the publication date in due course.

With kind regards,

Andrea Szabo

PLOS Genetics

On behalf of:
